# LET ME GROK FOR YOU: ACCELERATING GROKKING VIA EMBEDDING TRANSFER FROM A WEAKER MODEL

**Zhiwei Xu**[†*], **Zhiyu Ni**[‡*], **Yixin Wang**[† ◇], **Wei Hu**[† ◇]
[†]University of Michigan, [‡]University of California, Berkeley
`{zhiweixu,yixinw,vvh}@umich.edu,zhiyuni@berkeley.edu`

## ABSTRACT

"Grokking" (Power et al., 2022) is a phenomenon where a neural network first memorizes training data and generalizes poorly, but then suddenly transitions to near-perfect generalization after prolonged training. While intriguing, this delayed generalization phenomenon compromises predictability and efficiency. Ideally, models should generalize directly without delay. To this end, this paper proposes `GrokTransfer`, a simple and principled method for accelerating grokking in training neural networks, based on the key observation that data embedding plays a crucial role in determining whether generalization is delayed. `GrokTransfer` first trains a smaller, weaker model to reach a nontrivial (but far from optimal) test performance. Then, the learned input embedding from this weaker model is extracted and used to initialize the embedding in the target, stronger model. We rigorously prove that, on a synthetic XOR task where delayed generalization always occurs in normal training, `GrokTransfer` enables the target model to generalize directly without delay. Moreover, we demonstrate that, across empirical studies of different tasks, `GrokTransfer` effectively reshapes the training dynamics and eliminates delayed generalization, for both fully-connected neural networks and Transformers.

## 1 INTRODUCTION

"Grokking" is an intriguing phenomenon recently discovered by Power et al. (2022), where a neural network first memorizes the training dataset but has poor test performance, and after much longer training, it suddenly transitions to near-perfect generalization. Initially reported for Transformer models trained on modular arithmetic tasks, the grokking phenomenon has since been observed in other settings such as learning group operations (Chughtai et al., 2023), sparse parity (Barak et al., 2022), and image classification (Liu et al., 2023).

While grokking is an interesting phenomenon, it introduces unpredictability into the training process and compromises its practical efficiency. When the model has interpolated the training data with small training loss but still performed poorly on the validation set, it becomes difficult to predict whether or when the model will eventually generalize. Ideally, we would like the model to make continuous progress during training, keeping the gap between training and validation errors minimal. This raises the question:

*How can we effectively modify the training dynamics so that the model generalizes without delay?*

In this work, we show that data embedding plays a crucial role in determining the training dynamics; an informative embedding enables continuous progress during training. To obtain such an informative embedding without excessive computational cost, we propose a novel method called `GrokTransfer`, which leverages the embedding learned by a weaker, smaller model to accelerate the generalization of a larger target model. See Figure 1a for an overview of `GrokTransfer`.

Specifically, `GrokTransfer` involves two main steps: (1) Train a weaker model until it groks to non-trivial test performance; (2) Extract the weak model's learned embedding and use a linear

---

*Equal contribution; ◇ Equal advising.

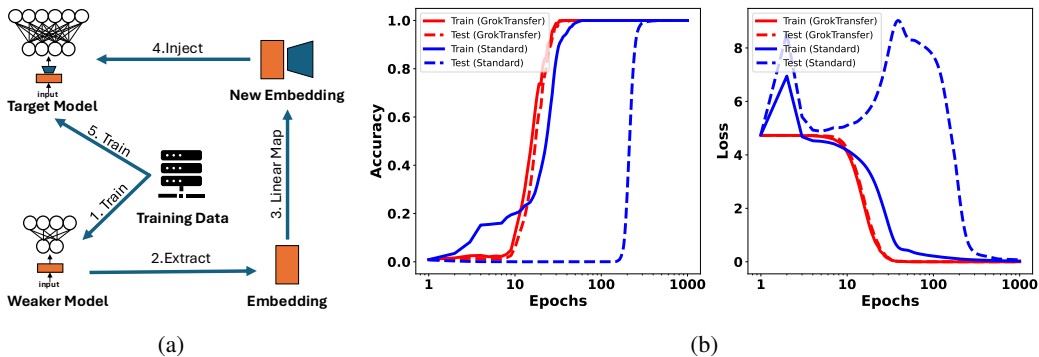

Figure 1: (a) Overview of the `GrokTransfer` framework. (b) Comparison of the training dynamics of a model trained using `GrokTransfer` versus one trained from scratch. There is a clear phase transition between memorization and generalization if we train the model from scratch (blue lines). `GrokTransfer` (red lines) enables the model to make continuous progress, significantly reducing the gap between memorization and generalization. See Appendix A.3 for the detailed experimental setup.

mapping of this embedding to initialize the embedding of the target model. Then, proceed to train the target model. We theoretically study `GrokTransfer` in the setting of a two-layer neural network trained on a high-dimensional XOR classification task, where normal training exhibits grokking. We prove that `GrokTransfer` enables the target model to directly generalize without delay. We further empirically verify the effectiveness of `GrokTransfer` on typical algorithmic tasks that show grokking. This is done for both fully-connected neural networks with trainable embeddings and Transformers. Figure 1b shows typical training curves of `GrokTransfer` vs. training a target model from scratch, on a modular addition task. It shows that `GrokTransfer` effectively eliminates grokking and significantly improves efficiency.

In summary, our contributions are as follows:

- We propose a novel method, `GrokTransfer`, which leverages the embedding learned from a smaller, weaker model to accelerate grokking in the target model.
- We theoretically justify `GrokTransfer` in an XOR classification task. We further empirically validate our method on several algorithmic tasks that exhibit grokking in normal training, demonstrating that `GrokTransfer` can effectively eliminate delayed generalization.

## 1.1 RELATED WORK

Our work draws on two themes around grokking and weak-to-strong knowledge transfer.

**Grokking.** Liu et al. (2022) reported that the model starts grokking when it learns the hidden structure of the data. Gromov (2023) showed that grokking is robust to different optimizers such as vanilla gradient descent and Adam; and regularization methods including no regularization, weight decay, and dropout. Davies et al. (2023) hypothesized that grokking and double descent, another surprising phenomenon, are caused by the same hidden mechanism. Nanda et al. (2023) reverse-engineered a grokked transformer model for modular addition and reported that the learned algorithm is a composition of trigonometric and inverse trigonometric functions. Merrill et al. (2023) and Varma et al. (2023) contributed to the occurrence of grokking to the competition of sparse (generalizing) and dense (complementary) subnetworks during training. Zhu et al. (2024) showed that models only grok when the training data exceeds some critical size. Liu et al. (2023) attributed grokking to large initialization scale and induced grokking on real-world datasets such as MNIST and IMDb by initializing models with large weight norm. Further work (Miller et al., 2023; Humayun et al., 2024) showed that grokking can also be observed in other scenarios such as Gaussian Process regression and multi-class classification with adversarial samples. A series of theoretical papers have established rigorous results for grokking/delayed generalization in several settings outside of algorithmic tasks: linear regression with linear models (Žunkovič & Ilievski, 2022), and binary classification with neural networks (Lyu et al., 2024; Xu et al., 2024). Lyu et al. (2024) proved that grokking can be induced by a sharp phase transition from kernel regime to rich regime. Mallinar et al. (2024) trained Recursive

Feature Machines on algorithmic tasks and found its training dynamics similar to neural networks, showing that grokking is not restricted to neural networks. He et al. (2024); Wang et al. (2024) found transformers achieve out-of-distribution generalization on some tasks through grokking. Doshi et al. (2024) provided analytical solutions for complex modular arithmetic tasks and hypothesized that some complex modular polynomial tasks cannot be learned by shallow neural networks. Mohamadi et al. (2024) showed that learning modular addition is fundamentally hard for neural networks in the kernel regime. A related phenomenon, termed "sudden drop in the loss" (Chen et al., 2024; Gopalani et al., 2025; Yang et al., 2025), describes an abrupt drop in loss after an extended plateau during online training.

Recent work has proposed several methods to accelerate grokking. Liu et al. (2023) explained grokking through the concept of a "Goldilocks zone", a spherical shell of weights, and found that restricting the weight norm to a sphere of the appropriate radius during training can accelerate generalization. However, this method introduces instability in the training process and still involves a phase transition. Furuta et al. (2024) suggested initializing the model with weights or embeddings from another model that has already generalized on a different task may accelerate grokking, which needs to train the same model on additional data, while our method do not need additional data. Lee et al. (2024) decomposed the gradient at each step and accelerated grokking by amplifying part of the gradient. Interestingly, Minegishi et al. (2024) demonstrated that the gap between memorization and generalization can be nearly eliminated if a lottery ticket, a set of sparse mask matrices, is applied to the model during training. However, this lottery ticket can only be obtained by first training the same model under the same initialization till generalization. In contrast, our approach can nearly eliminate the phase transition without requiring additional data or pretraining on the same model.

**Weak to strong knowledge transfer.**    Burns et al. (2023) proposed a method where a small model is first fine-tuned as a teacher model. This teacher model is then used to generate pseudo-labels to fine-tune a larger student model. Surprisingly, the student model can outperform the teacher. Wang et al. (2023) designed a learned linear growth operator, which uses a learnable linear map of a pretrained small model's weights as the initialization for the large model's weights, to accelerate the training of large models. In contrast to these works, our method focuses on transferring the embedding layer from a weaker model to the target model and reshaping the training dynamics to accelerate grokking.

## 1.2 NOTATION

For a set $S$ with finite elements, we denote its cardinality by $|S|$ and use Uniform$(S)$ to represent the uniform distribution over $S$. We denote the set $\{1, 2, \cdots, n\}$ by $[n]$. We use $\text{sgn}(x)$ to represent the sign of a scalar $x$. For a matrix $A \in \mathbb{R}^{n \times m}$, we denote by $A_{i,\cdot} = [A_{i,1}, \cdots, A_{i,m}]$ the $i$-th row, $A_{i:j,\cdot} = [A_{i,\cdot}^\top, \cdots, A_{j,\cdot}^\top]^\top \in \mathbb{R}^{(j-i+1) \times m}$ the $i$-th to $j$-th rows, and $\|A\|_\text{F}$ the Frobenius norm. We use $\phi(x) = \max\{0, x\}$ to represent the ReLU activation function. We denote the inner product between two vectors $a, b$ by $\langle a, b \rangle$. For two sequences $\{x_n\}$ and $\{y_n\}$, we say $x_n = O(y_n)$ if there exists some constant $C > 0$ such that $x_n \leq C y_n$ for all $n$ and $x_n = \Omega(y_n)$ if $y_n = O(x_n)$.

## 2 ACCELERATING GROKKING VIA EMBEDDING TRANSFER FROM A WEAKER MODEL

### 2.1 MOTIVATION: THE ROLE OF DATA EMBEDDING

To demonstrate the pivotal role of data embedding in shaping training dynamics, we examine the modular addition task $a + b \bmod p$. Following settings in Nanda et al. (2023) and Liu et al. (2023), we take $p = 113$. The dataset consists of $\{((a, b), y)\}_{0 \leq a, b \leq p-1}$ with label $y = (a + b) \bmod p$. 25% of the dataset is randomly sampled as the training set. We evaluate four types of embeddings:

- **One-hot embedding:** Each integer $a \in [0, p-1]$ is represented by its one-hot encoding.
- **Binary embedding:** Each $a$ is encoded in binary, padded with zeros to the maximum length $\lfloor \log_2(p-1) \rfloor + 1$.
- **Fourier embedding:**    Each $a$ is encoded as a vector of trigonometric functions: $[\cos(\frac{2\pi i_1 a}{p}), \sin(\frac{2\pi i_1 a}{p}), \cdots, \cos(\frac{2\pi i_k a}{p}), \sin(\frac{2\pi i_k a}{p})]$, where $i_1, \cdots, i_k \in \mathbb{N}$ are predetermined frequencies.

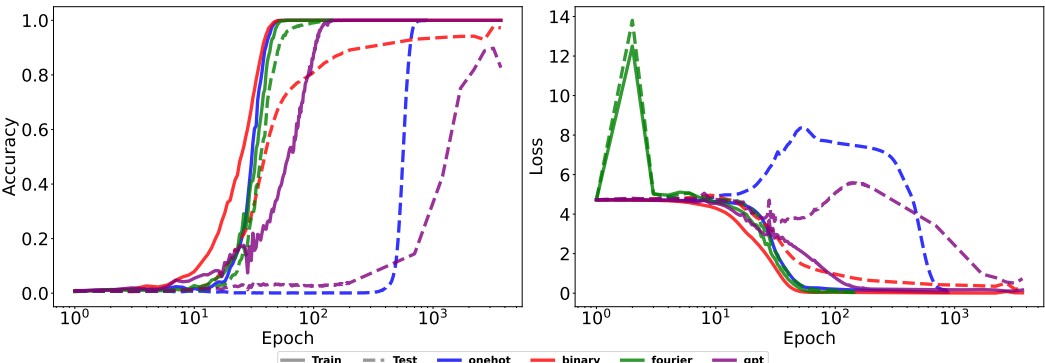

Figure 2: FNN training dynamics using different embeddings for the modular addition task ($p = 113$). The training dynamics vary significantly across different embeddings. The one-hot embedding and GPT embedding exhibit sharp phase transition. See Appendix A.3 for details of the experimental setup.

- **GPT embedding:** Each $a$ is embedded using OpenAI's `text-embedding-3-small` model (OpenAI, 2024)

One-hot embeddings contain no prior information about the data, while binary embeddings capture the ordinal information of integers. Fourier embeddings, inspired by the analytical solutions learned by neural networks (Nanda et al., 2023; Morwani et al., 2024), encode task-specific information. GPT embeddings encode general information about integers. Figure 2 shows the training dynamics of a feed-forward neural network using these embeddings. The training dynamics with one-hot and GPT embeddings exhibit clear grokking behavior, whereas those with binary and Fourier embeddings show continuous generalization progress. Notably, Fourier embeddings enable the model to simultaneously achieve memorization and perfect generalization. We observe that general embeddings like one-hot and GPT embeddings suffer from generalization delay, while embeddings encoded with task-related information allow the model to generalize continuously.

A series of works (Liu et al., 2023; Kumar et al., 2024; Lyu et al., 2024; Mohamadi et al., 2024) found that the default initialization scale is relatively large and causes generalization delay. They observed that reducing the initialization scale can accelerate grokking and hypothesized that grokking arises from a time gap between the Neural Tangent Kernel (NTK) regime and the feature-learning regime. However, our empirical findings indicate that grokking persists even after carefully tuning the initialization scale (see Appendix A.3.1). This suggests that grokking occurs even when the model is not initialized in the kernel regime, implying that the kernel regime may not be the sole cause of grokking. In Figure 3, we compare the changes in the empirical NTK (Mohamadi et al., 2023)

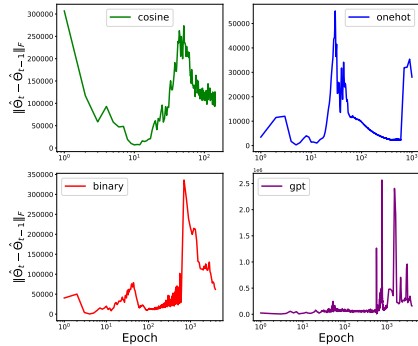

Figure 3: Change of empirical NTK.

corresponding to the dynamics in Figure 2. The change of empirical NTK evolves similarly across all four types of embeddings (see Appendix A.3 for details).

In conclusion, the choice of embedding significantly impacts training dynamics, and an informative embedding can close the gap between memorization and generalization. However, finding such an informative embedding for specific tasks is not always straightforward. Binary embedding, for example, reduces the sharp phase transition for modular addition but fails to do so for modular multiplication. In the next section, we will show that constructing a task-specific embedding from training data can be a promising approach to obtaining an informative embedding that can accelerate grokking. The embedding construction can be achieved by training a much smaller, weaker model. Here "small" refers to smaller model expressivity. This weaker model can learn an informative embedding without achieving optimal generalization. This embedding can then be used to positively influence the training dynamics of the larger target model.

## 2.2 OUR METHOD: GROKTRANSFER

We propose `GrokTransfer`, a simple and principled method for accelerating grokking in training neural networks. In more detail, given a specific task and a training set $\mathcal{G}$, we consider a target model $f_T$ that has a trainable embedding layer $E_T$ with vocabulary size $d_v$ and embedding dimension $d_T$. Our proposed method `GrokTransfer` works as follows:

1. **Train a Weaker Model:** Train a weaker model $f_W$ with a trainable embedding table $E_W \in \mathbb{R}^{d_v \times d_W}$ on $\mathcal{G}$, where $d_W$ is the embedding dimension in the weak model. Train $f_W$ until it groks to a non-trivial performance on the validation set.

2. **Train the Target Model:** Initialize $A = E_W$ and randomly initialize a matrix $B \in \mathbb{R}^{d_W \times d_T}$. Train the target model with an embedding layer set to $E_T = A \cdot B$, where both $A$ and $B$ are trainable.

By training a weaker model, the first step aims to obtain an informative embedding that aids the training of the target model. In practice, the weak model can be much smaller than the target model or can even have a different architecture (e.g., the weak model can be a fully-connected network when the target model is a Transformer; see Section 4). As a result, training a weak model greatly reduces the computational cost of acquiring an informative embedding. This contrasts with the method proposed in Minegishi et al. (2024), which requires the target model to be trained till perfect generalization first. In the next sections, we will demonstrate, both theoretically and empirically, that even if the weak model only partially generalizes (i.e., has a non-trivial but non-optimal test error), its embedding still allows the large model to generalize optimally without delay.

In the second step, we impose a low-rank structure $A \cdot B$ on the embedding $E_T$ while training the target model. This constraint alters the empirical risk landscape and provides a favorable initialization for the embedding table. The intuition behind our method is as follows: by initializing with an informative embedding from the weak model, the target model can bypass the initial phase of pure memorization. Instead, it can start generalizing almost immediately as it begins to optimize the training loss.

## 3 CASE STUDY: GROKTRANSFER ON XOR CLUSTER DATA

In this section, we theoretically study an XOR classification task and prove that `GrokTransfer` can eliminate grokking for this task.

### 3.1 THE SETUP OF XOR CLUSTER DATA

We study the setting where the data $x = [x_1, x_2, \cdots, x_p]^\top = [x_{\text{signal}}^\top, x_{\text{noise}}^\top]^\top \in \mathbb{R}^p, x_{\text{signal}} \sim$ Uniform($\{\pm 1\}^2$), $x_{\text{noise}} \sim$ Uniform($\{\pm \varepsilon\}^{p-2}$), and the label $y = x_1 x_2$. Here $\varepsilon$ is the parameter that controls the scale of the noise. We denote this data distribution by $P$ and consider $n$ training datapoints $\{(x_i, y_i)\}_{i=1}^n$ drawn i.i.d. from the distribution $P$. We assume the sample size $n$ to be sufficiently large, specifically larger than any universal constant mentioned in this paper. The data distribution comprises four feature vectors (see Figure 5a for a projected visualization), and the model need learn all four features to achieve perfect generalization.

We denote a width-$m$ two-layer neural network by $f(x) = \sum_{j=1}^m a_j \phi(\langle w_j, x \rangle)$, where $w_j \in \mathbb{R}^p, j \in [m]$ are neurons in the hidden layer and $a_j \in \mathbb{R}, j \in [m]$ are second-layer weights. The model is randomly initialized by

$$w_j \overset{i.i.d}{\sim} N(0, w_{\text{init}}^2 I_p), \quad a_j \overset{i.i.d}{\sim} N(0, a_{\text{init}}^2), \quad j \in [m].$$

Define the empirical risk with the exponential loss as: $\widehat{L}(f) = \sum_{i=1}^n l(y_i, f(x_i))/n$, where $l(y, \widehat{y}) = \exp(-y\widehat{y})$. We use gradient descent (GD) with weight decay $\theta_j^{(t+1)} = (1 - \lambda)\theta_j^{(t)} - \alpha \nabla_{\theta_j} \widehat{L}(f^{(t)})$ to update both layers $\{w_j, a_j\}_{j=1}^m$, where $\lambda$ is the coefficient of $L_2$ regularization.

Setting $p = 80000, n = 400, \varepsilon = 0.05$, this configuration approximates one of the distributions explored in Xu et al. (2024), where grokking was observed. Under this setup, we train a two-layer neural network on $\{(x_i, y_i)\}_{i=1}^n$ with default PyTorch initialization. We observe grokking, as shown in Figure 4(a), where overfitting is achieved by the fifth epoch and generalization begins around the

80-th epoch. Below we will show how our method `GrokTransfer` constructs a new embedding and eliminates the observed delay in generalization in subsequent sections.

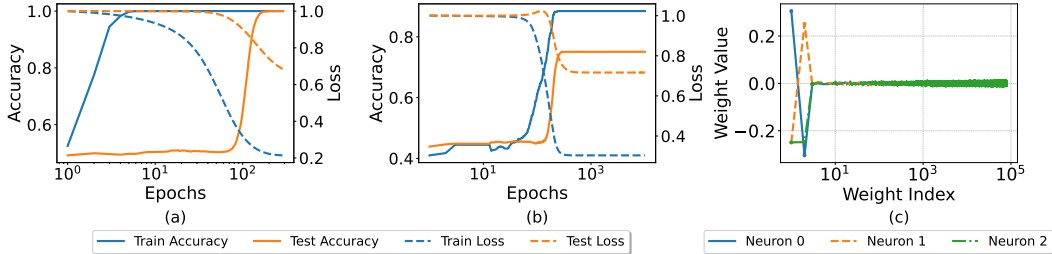

Figure 4: (a) Training dynamics of a two-layer neural network with a hidden width of $2048$, where grokking is observed. (b) Training dynamics of a two-layer neural network with a hidden width of 3. The model can only achieve around $75\%$ validation accuracy and a phase transition near 100th epoch is observed. (c) Visualization of individual neuron weights from the model trained in (b). It shows three distinct patterns and each corresponds to a feature direction of the XOR data distribution. See Appendix A.3 for details of the experimental setup.

## 3.2 EMPIRICAL ANALYSIS OF THE WEAKER MODEL

Applying `GrokTransfer`, we first train a small two-layer neural network with only 3 neurons $f_S(x) = \sum_{j=1}^{3} a_j \phi(\langle w_j, x \rangle)$ till convergence (Figure 4(b)). Denote the first-layer weight matrix by $W = [w_1, w_2, w_3] \in \mathbb{R}^{p \times 3}$, the number of training steps by $T$, and the model after training by $f_S^{(T)}$. Due to the complexity of the training dynamics, it is hard to derive the closed form of $f_S^{(T)}$ and $W^{(T)}$. Below we empirically investigate what information the model has gained and how well it learns.

Figure 4(b) shows that, after training, this weak model has non-trivial performance with test accuracy around $75\%$. The neurons $\{w_j^{(T)}\}_{j=1}^3$ are visualized in Figure 4(c), displaying patterns $[-1, 1, 0, \cdots, 0], [1, -1, 0, \cdots, 0]$, and $[-1, -1, 0, \cdots, 0]$. Note that the specific features learned by the model are sensitive to its initialization. Nevertheless, we find that empirically, the learned features are always three among the four features $[\pm 1, \pm 1, 0, \ldots, 0]$, provided the test accuracy is around $75\%$.

Notice that an optimal function for this classification task is

$$f(x) = \text{sgn}(\phi(x_1 + x_2) + \phi(-x_1 - x_2) - \phi(-x_1 + x_2) - \phi(x_1 - x_2)),$$

which needs four neurons to represent all features $[\pm 1, \pm 1]$. It thus follows intuitively that the weak model $f_S$ cannot achieve better generalization with only three neurons. Formally, we establish the following lemma regarding the expressive power of $f_S$.

**Lemma 3.1.** *For any $f(x) = \sum_{j=1}^{3} a_j \phi(w_j^\top x)$, where $\phi$ is the ReLU activation function, we have*

$$\mathbb{P}_{(x,y)\sim P}(y = \text{sgn}(f(x))) \leq 75\%.$$

Although the model $f_S^{(T)}$ fails to generalize perfectly due to the inherent limitation of capacity, it has correctly selected the subset that contains features after training as shown in Figure 4(c). Consequently, for any input $x \sim P$, $W^{(T)\top}x$ becomes a high-quality embedding for $x$ in a much lower dimensional space. Figure 5a shows that, with this new embedding, data points are well-separated in a three-dimensional space with a relatively high signal-to-noise ratio (SNR) compared to the original embedding.

Next, we empirically examine the order of the ratio between the norm of the complementary sub-network and the norm of the generalizing subnetwork. This will be used to estimate the SNR of $P$ with the new embedding. Given the structure of the XOR cluster data, the first two rows of $W^{(T)}$ correspond to the generalizing subnetwork. We define the norm ratio between the complementary

and generalizing subnetwork as follows:

$$r_W = \frac{\|W^{(T)}_{3:p,\cdot}\|_{\mathrm{F}}/\sqrt{p-2}}{\|W^{(T)}_{1:2,\cdot}\|_{\mathrm{F}}/\sqrt{2}}.$$

Figure 5b and 5c show that the norm ratio is proportional to $\varepsilon$ and $1/\sqrt{n}$, i.e. $r_W \propto \varepsilon/\sqrt{n}$. We will use this property to show that, under mild assumptions, the target model can learn this low-dimensional XOR task with just one step of gradient descent.

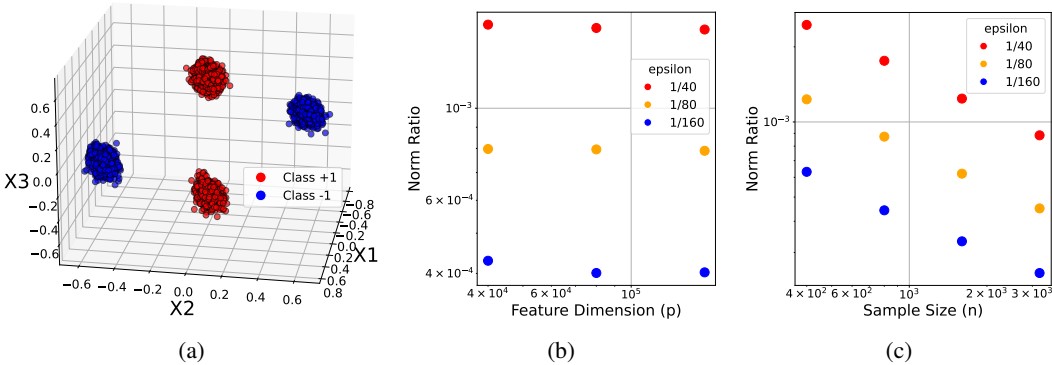

(a)           (b)           (c)

Figure 5: (a) 3-D Visualization of the distribution $P$ with the embedding from the weak model. The clusters are well-separated under the new embedding. (b) Norm ratio $r_W$ for different values of $p$ and $\varepsilon$ with fixed sample size $n$, indicating that $r_W$ does not depend on $p$. (c) Norm ratio $r_W$ for different values of $n$ and $\varepsilon$ with fixed feature dimension $p$. For each $\epsilon$, the slope is around $-1/2$, indicating that $r_W$ is proportional to $1/\sqrt{n}$. See Appendix A.3 for details of the experimental setup.

### 3.3 THEORETICAL ANALYSIS OF THE TARGET MODEL

In this section, we theoretically analyze the behavior of `GrokTransfer` on the XOR cluster data. We consider the target model as a large model with width $m$ of the form $f_L(x) = \sum_{j=1}^{m} a_j \phi(\langle v_j, U^\top x \rangle)$, where $U = [u_1, u_2, u_3] \in \mathbb{R}^{p \times 3}$ comes from the first-layer weight matrix $W^{(T)}$ learned by the weak model (visualized in Figure 4(c)). Here, $U$ is the embedding matrix being transferred from the weak model $f_S$, which will then go through another linear transformation (given by $v_j$'s) to form the embedding in the target model. Following our observation in Section 3.2, we can write

$$u_1 = [\mu_2^\top, \delta_1^\top]^\top, \quad u_2 = [-\mu_2^\top, \delta_2^\top]^\top, \quad u_3 = [-\mu_1^\top, \delta_3^\top]^\top,$$

where $\mu_1 = [1, 1]^\top, \mu_2 = [-1, 1]^\top$ are two orthogonal features of $P$, and $\delta_j = [\delta_{j,1}, \cdots, \delta_{j,p-2}]^\top \in \mathbb{R}^{p-2} (j \in [3])$.[1] Here we let $\delta = [\delta_1, \delta_2, \delta_3] = W^{(T)}_{3:p,\cdot}$. Given a universal constant $C > 1$, we assume

(A1) The noise scale $\varepsilon \le (n/(p \log^3 n))^{1/4}$.

(A2) The norm of the complementary subnetwork satisfies $\|\delta\|_{\mathrm{F}} \le C\varepsilon\sqrt{p/n}$.

(A3) The initialization scale $v_{\mathrm{init}} \le C \log^{-3/2}(n)$.

(A4) The step size $\sqrt{m}v_{\mathrm{init}}/C \le \alpha \le \sqrt{m}v_{\mathrm{init}}$.

(A5) The number of neurons satisfies $m \ge 2\log^3 n$.

Here Assumption (A2) corresponds to the finding that $r_W \propto \varepsilon/\sqrt{n}$ in Section 3.2. Assumptions (A1) and (A2) together ensure that the SNR of the distribution $P$ in the new embedding space is large enough. Assumption (A3) controls the initial weight norm of the target model such that the empirical risk starts within a reasonable range. Assumption (A4) guarantees that the step size is appropriately balanced; it is neither too small to prevent meaningful updates after a single-step gradient descent nor

---

[1]We assume that the weak model learned three features $[1, 1], [-1, 1], [-1, -1]$ without loss of generality. Our result will hold the same for any three features among the four features $[\pm 1, \pm 1]$.

too large to cause overly drastic movements. Assumption (A5) ensures that the model's width is large enough to ensure certain concentration results about the random initialization. All assumptions are satisfied in the empirical setup discussed in Section 3.2.

We denote $a = [a_1, \cdots, a_m]^\top \in \mathbb{R}^m$ and $V = [v_1, \cdots, v_m] \in \mathbb{R}^{3 \times p}$. We initialize $a$ and $V$ as follows:

$$a_j \overset{i.i.d}{\sim} \text{Uniform}(\{\pm 1/\sqrt{m}\}), \quad v_j \overset{i.i.d}{\sim} \text{Uniform}(\{\pm v_{\text{init}}\}^3), \quad j \in [m],$$

and keep $a$ and $U$ fixed during the training process.[2] Following the training method outlined in Section 3.1, we use gradient descent $V^{(t+1)} = V^{(t)} - \alpha \nabla_V \widehat{L}(f_L^{(t)})$ at step $t$ to update the linear layer $V$, where $\alpha$ is the step size and the empirical risk $\widehat{L}(\cdot)$ is defined in Section 3.1. With the assumptions and initializations, we state the theorem that characterizes the train and test error of the target model after one step.

**Theorem 3.2.** *Suppose that Assumptions (A1)-(A5) hold. With probability at least $1 - O(1/n^2)$ over the generation of the training data and initial weights of $f_L$, after one step of training, the classifier* $\text{sgn}(f_L^{(1)}(x))$ *can correctly classify all training datapoints and generalize with test error no greater than* $\exp(-\Omega(\log^2 n))$.

Theorem 3.2 shows that with `GrokTransfer`, after just one step of gradient descent, the target model overfits all training data and achieves near perfect test accuracy. Notably, this is not in a kernel regime but a feature learning regime. Since models with normal training cannot achieve generalization in one step (Figure 4(a)), this result indicates that our method `GrokTransfer` effectively boosts the generalization speed of the target model and eliminates the time gap between overfitting and generalization. Empirically, the model continues to generalize with further training (see Figure 9 in Appendix A.3). Given that the weaker model $f_S$ has only three neurons, the computational cost of training $f_S$ is negligible compared to the cost of training the target model $f_L$ with sufficiently large width. This implies that `GrokTransfer` may reduce the overall computational cost. In the next section, we will compare the computational cost of our method to that of standard training procedures.

## 4 EXPERIMENTS

This section empirically studies `GrokTransfer` in modular addition and multiplication, as well as the sparse parity task. Our experiments verify that `GrokTransfer` effectively reshapes the training dynamics and eliminate delayed generalization for both fully-connected neural networks (FNN) and Transformers (TF). The AdamW optimizer (Loshchilov & Hutter, 2019) is used in all experiments in this section.

### 4.1 FNN → FNN

We first consider a three-layer FNN as the target model and conduct `GrokTransfer` on tasks including modular addition, modular multiplication, and $(q, k)$-parity (Barak et al., 2022). These results are compared to training a target model from scratch. The modular addition task is introduced in Section 2.1, and modular multiplication is defined similarly with the label $y = ab \bmod p$. The $(q, k)$-parity task consists of a dataset $\{(x, y) : x \in \{\pm 1\}^q, y = \prod_{i \in S} x_i, |S| = k\}$. Following the setting in Merrill et al. (2023), we choose $q = 40, k = 3$, and $S = \{1, 2, 3\}$.

For the modular addition and multiplication tasks, we employ a two-layer neural network with a trainable embedding as the weak model, which we train for $10^4$ epochs. We then initialize the target model by setting its layer $A$ to the embedding learned by the weak model. Figure 6a and 6b show the training dynamics of the weak model, the target model trained via `GrokTransfer`, and the target model trained from scratch. Notably, `GrokTransfer` nearly eliminates the sharp phase transition observed in normal training. Here all training hyperparameters (initialization scale, learning rate, weight decay) are selected by grid search, and the best configuration is defined as the one that reaches $99\%$ test accuracy the quickest. The oscillations of accuracies in the second row of Figure 6 are related to the "slingshot mechanism" (Thilak et al., 2022) and training instabilities associated with large learning rates (Wortsman et al., 2024). Since large learning rate and this kind of oscillation

---

[2]Our result will not be affected if $a$ and $U$ are also trainable. We set them fixed to simplify the analysis while still conveying the main ideas.

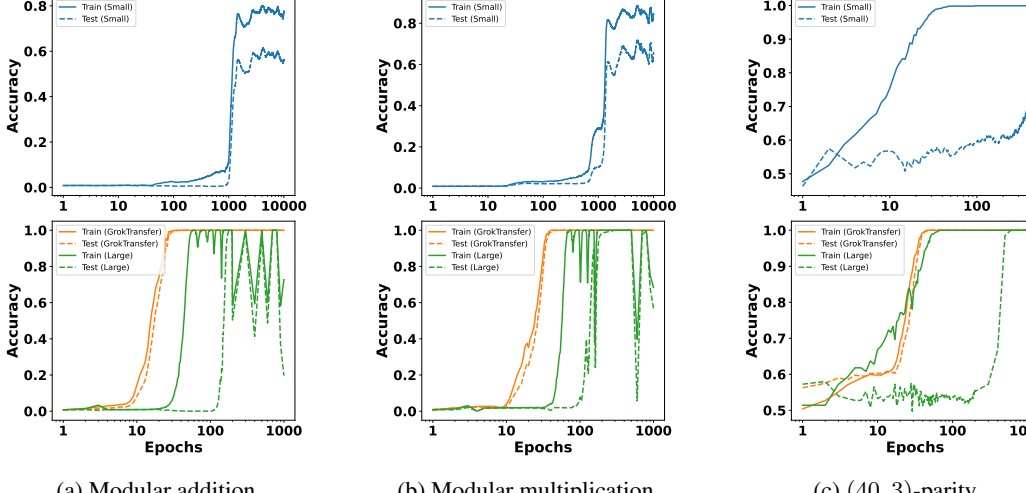

(a) Modular addition  (b) Modular multiplication  (c) $(40, 3)$-parity

Figure 6: Training dynamics of FNNs on various tasks. The rows represent different models/training methods: The first row shows the dynamics of the weak model used in `GrokTransfer`, the second row shows the dynamics of the target model trained using `GrokTransfer`, and the target model trained from scratch. The columns represent different tasks: the first column is for the modular addition task, the second column is for the modular multiplication task, and the third column is for the $(40, 3)$-parity task. The comparison between the first and second rows shows that the target model trained via `GrokTransfer` can surpass the weak model's performance. The comparison within the second row shows that `GrokTransfer` eliminates the sharp phase transition and enables the model to make continuous progress. See Appendix A.3 for details of the experimental setup.

are believed to help generalization (Damian et al., 2023; Lu et al., 2024), we do not change our configuration selection criteria.

For the parity task, we use a three-layer FNN as the weak model, as empirical evidence suggests that a two-layer FNN without bias terms cannot generalize on this task. The weak model is trained until it achieves $70\%$ test accuracy, after which the first layer's weight matrix is transferred to the target model. As shown in Figure 6c, the weak model undergoes a generalization delay, but the large model inheriting its embedding generalizes continuously.

**Ablation study:** To further understand the empirical effectiveness of `GrokTransfer`, we perform an ablation study by varying the training epochs of the weak model in the modular addition task. We extract the embeddings of the weak model at epochs $100, 500, 800, 900, 1000, 1100, 1500$, and $2000$. For each embedding, we apply `GrokTransfer` to the target model and train it for $10^4$ epochs. To measure the generalization delay of the target model, we define *Time Gap* as the difference between the first epoch that achieves $95\%$ training accuracy and the first epoch that achieves $95\%$ test accuracy. If the target model fails to reach $95\%$ accuracy, we set $1/Time\ Gap = 0$. Figure 7 shows that the test performance of the target model, initialized with the weak model's embedding, is positively correlated with the test performance of the weak model. A grokked weak model is essential for the target model to achieve near-perfect generalization with minimal generalization delay. We hypothesize that the target model can only generalize well after the weak model has grokked.

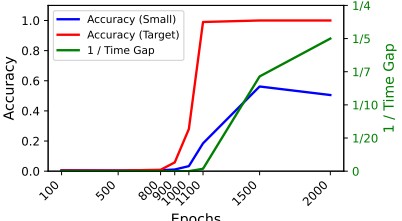

Figure 7: Ablation study showing the effect of the weak model's performance on the test accuracy of the target model (initialized via `GrokTransfer` and trained for $10^4$ epochs).

### 4.2 FNN → TRANSFORMERS

Interestingly, we find that the embeddings extracted from the weak FNN model can be transferred to the target model even when the target model is a Transformer comparable to the scale of GPT2-small (Radford et al., 2019). Under this FNN → TF setting, `GrokTransfer` still mitigates the generalization delay of the target model. Specifically, we choose the target model to be a Transformer

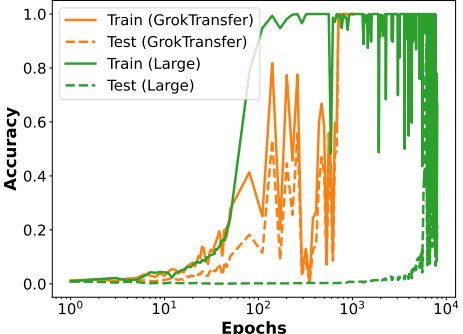 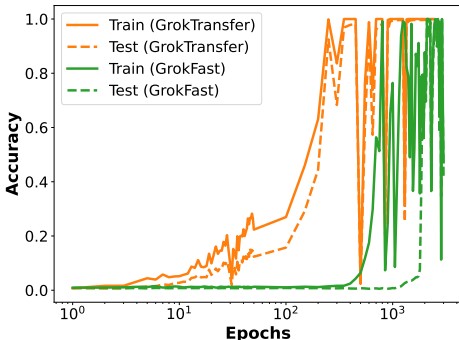

Figure 8: Training dynamics of Transformers on Modular Addition Task. The weak model is a three-layer FNN. (a) Dynamics of the target model (an 8-layer transformer) trained via `GrokTransfer`, and the target model trained from scratch. (b) Dynamics of the target model (a two-layer Transformer) trained via `GrokTransfer`, and the target model trained via `GrokFast` (Lee et al., 2024).

with 8 attention layers, $(d_{\text{embed}}, d_{\text{mlp}}, n_{\text{head}}) = (512, 512, 4)$. For each sample, the input is a sequence with two tokens $(a, b)$. We extract the embeddings of the weak model in Figure 6a at the point that it first reaches $30\%$ test accuracy. Figure 8(a) shows that `GrokTransfer` enables the target model to generalize much faster than training from scratch and exhibits little generalization delay. Here both method suffer from training instability of large learning rates.

In terms of the computation cost, we use wall-clock time as the measure. The computation cost of `GrokTransfer` comprises the training of weak model and the training of target model. Table 1 shows the total wall-clock time for weak model, target model with `GrokTransfer`, and target model trained from scratch. The time spent training the weak FNN model is negligible compared to training the target transformer model. The total wall-clock time of `GrokTransfer` is approximately five times faster than training from scratch.

| Model | Weak | Target (`GrokTransfer`) | Target (scratch) |
|---|---|---|---|
| Total Wall-clock time (ms) | 2828 | 71079 | 392667 |

Table 1: Comparison of total wall clock times (forward and backward passes) for different models. The weak model is a three-layer FNN. The target/large model is an 8-layer transformer.

Lee et al. (2024) proposed a gradient amplification algorithm `GrokFast` to accelerate grokking. We compare `GrokTransfer` with `GrokFast` in Figure 8(b). The weak model embedding we transfer is the same as the one used in Figure 8(a). For the target model, we follow the model used in Lee et al. (2024), which is a two-layer decoder-only transformer with $(d_{\text{embed}}, d_{\text{mlp}}, n_{\text{head}}) = (128, 512, 4)$. The *Time Gap* of `GrokTransfer` is 46 while the *Time Gap* of `GrokFast` is 1119.

## 5 CONCLUSION

To eliminate the unpredictability associated with grokking, we proposed `GrokTransfer`, a novel method that effectively accelerates grokking by transferring the embedding from a weaker model. Our method was inspired by the key observation that data embedding critically shapes training dynamics. We theoretically justified `GrokTransfer` on an XOR classification task. We also empirically evaluated it on various algorithmic tasks known to exhibit grokking under standard training. Our results showed that `GrokTransfer` can effectively modify training dynamics, enabling continuous progression in model performance.

One limitation of our work is that the theoretical result only considers a relatively simple XOR task. For this task, after transferring the embedding from the smaller model, one step of gradient descent suffices for both memorization and generalization. Theoretical justification for more complex problems is an important future direction. Furthermore, our method focuses solely on accelerating grokking and was only investigated on problems where grokking occurs. It would be interesting to study whether similar ideas can be applied to improve training dynamics or enable weak-to-strong generalization in a broader context.

ACKNOWLEDGMENTS

This work was supported in part by the Office of Naval Research under grant number N00014-23-1-2590, the National Science Foundation under Grant No. 2231174, No. 2310831, No. 2428059, No. 2435696, No. 2440954, and a Michigan Institute for Data Science Propelling Original Data Science (PODS) grant.

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

# A    APPENDIX

## A.1    PROOFS

### A.1.1    PROOF OF LEMMA 3.1

**Lemma 3.1.** *For any $f(x) = \sum_{j=1}^{3} a_j \phi(w_j^\top x)$, where $\phi$ is the ReLU activation function, we have*

$$\mathbb{P}_{(x,y)\sim P}(y = \text{sgn}(f(x))) \leq 75\%.$$

*Proof.* For any $(x, y) \sim P$, define $x' = (x_1, -x_2, x_3, \cdots, x_p)$ and $y' = \text{sgn}(x_1' x_2') = -y$. It is sufficient to show that if $y = \text{sgn}(f(x)), y = \text{sgn}(f(-x)), y' = \text{sgn}(f(x'))$, then $y' \neq \text{sgn}(f(-x'))$ with probability 1.

Assume $y = \text{sgn}(f(x))$ and $y = \text{sgn}(f(-x))$. Given $\phi(z) \geq 0, \forall z$, $y = \text{sgn}(f(x))$ implies that there exists at least one $i \in [3]$ such that $a_i$ has the same sign as $y$ and $w_i^\top x > 0$. Without loss of generality, assume $\text{sgn}(a_1) = y, w_1^\top x > 0$. Then for $(-x, -y)$, it follows that

$$f(-x) = \sum_{j=1}^{3} a_j \phi(-w_j^\top x) = a_2 \phi(-w_2^\top x) + a_3 \phi(-w_3^\top x)$$

has the same sign as $y$. Again without loss of generality, we assume $\text{sgn}(a_2) = y$.

If $y' = \text{sgn}(f(x'))$ and $y' \neq \text{sgn}(f(-x'))$ hold, following the same discussion, we have that at least two $a_i$'s have the same sign as $y' = -y$, which contradicts the previous assumption that $\text{sgn}(a_1) = \text{sgn}(a_2) = y$. $\square$

### A.1.2    PROOF OF THEOREM 3.2

**Additional notations:**    For training dataset $\{(x_i, y_i)\}_{i=1}^{n}$, we denote the signal of $x_i$ by $\bar{x}_i = [x_{i,1}, x_{i,2}]^\top \in \{\pm\mu_1, \pm\mu_2\}$. For each $\mu \in \{\pm\mu_1, \pm\mu_2\}$, define

$$\mathcal{I}_\mu = \{i \in [n] : \bar{x}_i = \mu\}$$

and $n_\mu = |\mathcal{I}_\mu|$. Denote the new embedding of the $i$-th datapoint by $z_i = U^\top x_i, i \in [n]$. Define

$$\nu_i = [\mu_2, -\mu_2, -\mu_1]^\top \mu_i, \quad i = 1, 2.$$

Then $\nu_1 = [0, 0, -2], \nu_2 = [2, -2, 0]$, and $\{\pm\nu_1, \pm\nu_2\}$ becomes the features for $P$ with the new embedding. Denote the signal of $z_i$ by $\bar{z}_i = [\mu_2, -\mu_2, -\mu_1]^\top \bar{x}_i$. Define the set of training data

$$\mathcal{G}_{\text{data}} = \{\{(x_i, y_i)\}_{i=1}^n : \|z_i - \bar{z}_i\| \le \varepsilon^2 \sqrt{\frac{p}{n}} \log n, \text{ for all } i \in [n]\}.$$

By Lemma A.2, $\mathbb{P}(\{(x_i, y_i)\}_{i=1}^n \in \mathcal{G}_{\text{data}}) \ge 1 - \exp(-\Omega(\log^2 n))$. We further define sets to separate the second-layer coefficients for the ease of discussion:

$$\mathcal{J}_{\text{Pos}} = \{j \in [m] : a_j > 0\}; \quad \mathcal{J}_{\text{Neg}} = \{j \in [m] : a_j < 0\}.$$

We divide the index of neurons by its initialization and define $\mathcal{J}_e = \{j \in [m] : v_j^{(0)} = v_{\text{init}} e\}$ for $e \in \text{Uniform}(\{\pm 1\})^3$. We further define

$$\mathcal{J}_{\text{Pos},e} = \mathcal{J}_{\text{Pos}} \cap \mathcal{J}_e; \quad \mathcal{J}_{\text{Neg},e} = \mathcal{J}_{\text{Neg}} \cap \mathcal{J}_e.$$

For each initialization of $v_j^{(0)}$, we denote the set of datapoints which have positive inner product with it by

$$\mathcal{I}_{e,\mu} = \{i \in \mathcal{I}_\mu : \langle e, z_i \rangle > 0\}, \quad e \in \text{Uniform}(\{\pm 1\}^3), \mu \in \{\pm\mu_1, \pm\mu_2\}.$$

**Theorem 3.2.** *Suppose that Assumptions (A1)-(A5) hold. With probability at least $1 - O(1/n^2)$ over the generation of the training data and initial weights of $f_L$, after one step of training, the classifier $\text{sgn}(f_L^{(1)}(x))$ can correctly classify all training datapoints and generalize with test error no greater than $\exp(-\Omega(\log^2 n))$.*

*Proof.* For brevity, we omit the subscript $L$ in $f_L$ in the proof below.

At step $t = 0$: for each $(x_i, y_i)$, we have

$$f^{(0)}(x_i) = \sum_{j=1}^m a_j \phi(\langle v_j^{(0)}, z_i \rangle),$$

where $a_j \phi(\langle v_j^{(0)}, z_i \rangle), j \in [m]$ are bounded random variables with zero mean. The absolute bound is

$$|a_j \phi(\langle v_j^{(0)}, z_i \rangle)| \le \frac{\sqrt{3} v_{\text{init}}}{\sqrt{m}} (\max_i \|\bar{z}_i\| + \varepsilon^2 \sqrt{p/n} \log n) \le 5 v_{\text{init}}/\sqrt{m},$$

where the first inequality uses Lemma A.2 and the second inequality uses $\max_i \|\bar{z}_i\| = 2\sqrt{2}$ and Assumption (A1). Then by Hoeffding's inequality and law of total probability,

$$\mathbb{P}(|f^{(0)}(x_i)| > t) \le \mathbb{P}(|f^{(0)}(x_i)| > t | \mathcal{G}_{\text{data}}) + \mathbb{P}(\mathcal{G}_{\text{data}}) \le 2\exp\left(-\frac{2t^2}{25 v_{\text{init}}^2}\right) + \exp(-\Omega(\log^2 n)).$$

Let $t = v_{\text{init}} \log n$. It follows that

$$\mathbb{P}(\max_{i \in [n]} |f^{(0)}(x_i)| \le t) \ge 1 - \sum_{i=1}^n \mathbb{P}(|f^{(0)}(x_i)| > t)$$

$$\ge 1 - 2n \exp(-\frac{2 \log^2 n}{25}) - n \exp(-\Omega(\log^2 n)) = 1 - \exp(-\Omega(\log^2 n)). \tag{1}$$

We define a set of training data and initial weights:

$$\mathcal{G} = \Big\{ (\{(x_i, y_i)\}_{i=1}^n, a, V^{(0)}) : \{(x_i, y_i)\}_{i=1}^n \in \mathcal{G}_{\text{data}}, \text{condition (1) and}$$

$$\text{all conditions in Lemma A.1 and A.4 hold} \Big\}.$$

Combining (1), Lemma A.1, A.2, and A.4 then applying the union bound, we have

$$\mathbb{P}((\{(x_i, y_i)\}_{i=1}^n, a, V^{(0)}) \in \mathcal{G}) \ge 1 - \exp(-\Omega(\log^2 n)) - O(\frac{1}{n^2}) - O(\frac{1}{n^4}) = 1 - O(\frac{1}{n^2}).$$

Denote $l_i^{(t)} = l(y_i, f^{(t)}(x_i)) = \exp(-y_i f^{(t)}(x_i))$. Conditioning on $\mathcal{G}$, the ratio between the maximum and minimum loss is bounded by:

$$R^{(0)} := \frac{\max_{i \in [n]} l_i^{(0)}}{\min_{i \in [n]} l_i^{(0)}} \leq \exp(2v_{\text{init}} \log n). \tag{2}$$

For each $j$, below we will analyze the gradient descent update for all possible combinations of $a_j^{(0)}, v_j^{(0)}$ conditioning on the event $\mathcal{G}$.

**(1)** When $a_j > 0$: If $v_j^{(0)} = v_{\text{init}}[1, 1, 1]$, then according to Lemma A.1, we have

$$\mathcal{I}_{[1,1,1],+\mu_1} = \varnothing; \quad \mathcal{I}_{[1,1,1],-\mu_1} = \mathcal{I}_{-\mu_1}; \quad \left| |\mathcal{I}_{[1,1,1],\mu}| - \frac{n_\mu}{2} \right| \leq \sqrt{n \log n}, \mu = \pm\mu_2. \tag{3}$$

Recall that the gradient descent update of $v_j^{(t)}$ is

$$v_j^{(t+1)} = v_j^{(t)} + \frac{\alpha}{n} a_j \sum_{i=1}^n y_i \exp(-y_i f^{(t)}(x_i)) \phi'(\langle v_j^{(t)}, z_i \rangle) z_i. \tag{4}$$

It follows that

$$
\begin{aligned}
v_{j,3}^{(1)} &= v_{j,3}^{(0)} + \frac{\alpha}{n} a_j \sum_{i=1}^n y_i l_i^{(0)} \phi'(\langle v_j^{(0)}, z_i \rangle) z_{i,3} \\
&= v_{j,3}^{(0)} + \frac{\alpha}{n} a_j \sum_{i \in \mathcal{I}_{-\mu_1}} y_i l_i^{(0)} z_{i,3} + \frac{\alpha}{n} a_j \sum_{i \in \mathcal{I}_{[1,1,1],\mu_2} \cup \mathcal{I}_{[1,1,1],-\mu_2}} y_i l_i^{(0)} z_{i,3} \\
&\geq v_{\text{init}} + \frac{2\alpha}{n\sqrt{m}} \sum_{i \in \mathcal{I}_{-\mu_1}} l_i^{(0)} - O\left(\frac{\alpha}{\sqrt{m}} \max_i l_i^{(0)} \varepsilon^2 \sqrt{\frac{p}{n}} \log n\right) \\
&\geq v_{\text{init}} + \frac{1.9\alpha |\mathcal{I}_{-\mu_1}|}{n\sqrt{m}} \exp(-v_{\text{init}} \log n) \geq v_{\text{init}} + \frac{1.9\alpha}{4\sqrt{m}}\left(1 - \frac{4}{\log n}\right)(1 - v_{\text{init}} \log n) \\
&\geq v_{\text{init}} + \frac{2\alpha}{5\sqrt{m}},
\end{aligned} \tag{5}
$$

where the first inequality uses Lemma A.2 and $\bar{z}_{i,3} = 0, i \in \mathcal{I}_{\pm\mu_2}$; the second inequality uses Assumption (A1), (A3) and (A4); the third inequality uses Lemma A.4 and $\exp(x) \geq 1 + x$. Further for $l = 1, 2$, we have

$$
\begin{aligned}
\left| v_{j,l}^{(1)} - v_{j,l}^{(0)} \right| &= \left| \frac{\alpha}{n} a_j \sum_{i \in \mathcal{I}_{-\mu_1}} y_i l_i^{(0)} z_{i,l} + \frac{\alpha}{n} a_j \sum_{i \in \mathcal{I}_{[1,1,1],\mu_2} \cup \mathcal{I}_{[1,1,1],-\mu_2}} y_i l_i^{(0)} z_{i,l} \right| \\
&= \frac{\alpha}{n} a_j \Bigg| \sum_{i \in \mathcal{I}_{-\mu_1} \cup \mathcal{I}_{[1,1,1],\mu_2} \cup \mathcal{I}_{[1,1,1],-\mu_2}} y_i l_i^{(0)} (z_{i,l} - \bar{z}_{i,l}) \\
&\qquad - \Bigg[ \sum_{i \in \mathcal{I}_{[1,1,1],\mu_2}} l_i^{(0)} \bar{z}_{i,l} + \sum_{i \in \mathcal{I}_{[1,1,1],-\mu_2}} l_i^{(0)} \bar{z}_{i,l} \Bigg] \Bigg| \\
&\leq \frac{\alpha}{\sqrt{m}} \exp(v_{\text{init}} \log n) \varepsilon^2 \sqrt{\frac{p}{n}} \log n + \frac{2\alpha}{n\sqrt{m}} \Bigg| \sum_{i \in \mathcal{I}_{[1,1,1],\mu_2}} l_i^{(0)} - \sum_{i \in \mathcal{I}_{[1,1,1],-\mu_2}} l_i^{(0)} \Bigg| \\
&\leq \frac{\alpha}{\sqrt{m}} \exp(v_{\text{init}} \log n) \varepsilon^2 \sqrt{\frac{p}{n}} \log n + \frac{2\alpha}{n\sqrt{m}} \exp(v_{\text{init}} \log n) \Big( \frac{n}{8} + \frac{n}{2 \log n} \\
&\qquad + \sqrt{n \log n} - \exp(-2v_{\text{init}} \log n)\big(\frac{n}{8} - \frac{n}{2 \log n} - \sqrt{n \log n}\big) \Big) \\
&\leq C \frac{\alpha \varepsilon^2 \sqrt{p}}{\sqrt{mn}} \log n + C \frac{\alpha}{n\sqrt{m}} \Big( \frac{n}{\log n} + v_{\text{init}} n \log n \Big) \leq C \frac{\alpha}{\sqrt{m \log n}},
\end{aligned} \tag{6}
$$

where the first inequality uses (2) and $\bar{z}_{i,l} = -\bar{z}_{j,l}$ for $i \in \mathcal{I}_{[1,1,1],\mu_2}, j \in \mathcal{I}_{[1,1,1],-\mu_2}$; the second inequality uses (2), (3), and (B4) in Lemma A.4; the third inequality uses Assumption (A1)-(A5); and the last inequality uses Assumption (A1), (A3) and (A4).

For a datapoint $(x, y) \sim P$, define $z = [z_1, z_2, z_3]^\top = U^\top x$. Applying Lemma A.3 we obtain

$$\mathbb{P}(\|z - \bar{z}\| \leq \varepsilon^2 \sqrt{\frac{p}{n}} \log n) \geq 1 - \exp(-\Omega(\log^2 n)).$$

Conditioning on

$$\|z - \bar{z}\| \leq \varepsilon^2 \sqrt{\frac{p}{n}} \log n, \tag{7}$$

if $x_{\text{signal}} = -\mu_1$, we combine (5) and (6) and have

$$\langle v_j^{(1)}, z \rangle = \langle v_j^{(1)}, \bar{z} \rangle + \langle v_j^{(1)}, z - \bar{z} \rangle \geq 2(v_{\text{init}} + \frac{2\alpha}{5\sqrt{m}}) - C v_{\text{init}} \varepsilon^2 \sqrt{\frac{p}{n}} \log n \geq \frac{3}{2}(v_{\text{init}} + \frac{2\alpha}{5\sqrt{m}}). \tag{8}$$

Further for any pair $j_1, j_2$ with $v_{j_1}^{(0)} = v_{j_2}^{(0)} = v_{\text{init}}[1, 1, 1]$ and $a_{j_1} > 0, a_{j_2} < 0$:

If $\langle v_{j_2}^{(1)}, z \rangle < 0$, it follows that

$$z_3 \frac{\alpha}{n\sqrt{m}} \sum_{i=1}^n y_i l_i^{(0)} \phi'(\langle v_{j_2}^{(0)}, z_i \rangle) z_{i,3} = -\langle v_{j_2}^{(1)}, z \rangle + z_3 v_{j_2,3}^{(0)} + \sum_{l=1}^2 z_l v_{j_1,l}^{(1)}$$

$$\geq z_3 v_{\text{init}} - \sum_{l=1}^2 |z_l - \bar{z}_l| |v_{j_2,l}^{(1)}| \tag{9}$$

$$\geq z_3 v_{\text{init}} - 2\varepsilon^2 \sqrt{\frac{p}{n}} \log n (v_{\text{init}} + C \frac{\alpha}{\sqrt{m \log n}}) \geq \frac{z_3}{2} v_{\text{init}},$$

where the first inequality uses $v_{j_2,3}^{(0)} = v_{\text{init}}$ and $\bar{z}_l = 0, l = 1, 2$; the second inequality uses (7); and the last inequality uses condition (7), $\bar{z}_3 = 2$, and Assumption (A1), (A3) and (A4). Combining (5) and (9), we have

$$\frac{\alpha}{n\sqrt{m}} \sum_{i=1}^n y_i l_i^{(0)} \phi'(\langle v_{j_2}^{(0)}, z_i \rangle) z_{i,3} \geq \max\{\frac{v_{\text{init}}}{2}, \frac{2\alpha}{5\sqrt{m}}\},$$

which together with (6) yield that

$$a_{j_1} \phi(\langle v_{j_1}^{(1)}, z \rangle) + a_{j_2} \phi(\langle v_{j_2}^{(1)}, z \rangle) = a_{j_1} \langle v_{j_1}^{(1)}, z \rangle$$

$$= \frac{1}{\sqrt{m}} \left[ \langle v_{j_1}^{(0)}, z \rangle + \frac{\alpha}{n\sqrt{m}} z_3 \sum_{i=1}^n y_i l_i^{(0)} \phi'(\langle v_{j_2}^{(0)}, z_i \rangle) z_{i,3} + \sum_{l=1}^2 (v_{j_1,l}^{(1)} - v_{j_1,l}^{(0)}) z_l \right]$$

$$\geq \frac{1}{\sqrt{m}} \left[ v_{\text{init}} + \max\{\frac{v_{\text{init}}}{2}, \frac{2\alpha}{5\sqrt{m}}\} - C v_{\text{init}} \varepsilon^2 \sqrt{\frac{p}{n}} \log n - C \frac{\alpha}{\sqrt{m \log n}} \varepsilon^2 \sqrt{\frac{p}{n}} \log n \right] \tag{10}$$

$$\geq \frac{1}{\sqrt{m}} \left[ v_{\text{init}} + \frac{v_{\text{init}}}{4} + \frac{\alpha}{5\sqrt{m}} - C v_{\text{init}} \varepsilon^2 \sqrt{\frac{p}{n}} \log n - C \frac{\alpha}{\sqrt{m \log n}} \varepsilon^2 \sqrt{\frac{p}{n}} \log n \right]$$

$$\geq \frac{v_{\text{init}}}{\sqrt{m}} + \frac{\alpha}{10m},$$

where the second inequality uses $\max(x, y) \geq (x + y)/2$ and the last inequality uses the fact that $n$ is sufficiently large.

If $\langle v_{j_2}^{(1)}, z \rangle > 0$, we have

$$a_{j_1} \phi(\langle v_{j_1}^{(1)}, z \rangle) + a_{j_2} \phi(\langle v_{j_2}^{(1)}, z \rangle) = \frac{1}{\sqrt{m}} \langle v_{j_1}^{(1)} - v_{j_2}^{(1)}, z \rangle$$

$$= \frac{1}{\sqrt{m}} \langle v_{j_1}^{(1)} - v_{j_1}^{(0)}, z \rangle - \frac{1}{\sqrt{m}} \langle v_{j_2}^{(1)} - v_{j_2}^{(0)}, z \rangle$$

$$= \frac{1}{\sqrt{m}} \left[ 2 \frac{\alpha}{n\sqrt{m}} z_3 \sum_{i=1}^n y_i l_i^{(0)} \phi'(\langle v_{j_1}^{(0)}, z_i \rangle) z_{i,3} + \sum_{l=1}^2 (v_{j_1,l}^{(1)} - v_{j_2,l}^{(1)})(z_l - \bar{z}_l) \right] \tag{11}$$

$$\geq \frac{1}{\sqrt{m}} \left[ \frac{4\alpha}{5\sqrt{m}} - C \frac{\alpha}{\sqrt{m \log n}} \varepsilon^2 \sqrt{\frac{p}{n}} \log n \right] \geq \frac{2\alpha}{5m},$$

where the second equation uses $v_{j_1}^{(0)} = v_{j_2}^{(0)}$; the third equation uses (4); the first inequality uses (5); and the second inequality uses Assumption (A1). Combining (10) and (11), it follows that

$$a_{j_1}\phi(\langle v_{j_1}^{(1)}, z\rangle) + a_{j_2}\phi(\langle v_{j_2}^{(1)}, z\rangle) \geq \frac{2\alpha}{5m} \tag{12}$$

when $v_{j_1}^{(0)} = v_{j_2}^{(0)} = v_{\text{init}}[1, 1, 1]$ and $x_{\text{signal}} = -\mu_1$.

If $x_{\text{signal}} = +\mu_1$, following the same procedure, we obtain that $\langle v_j^{(1)}, z\rangle < 0$ for $a_j > 0$. For $a_j < 0$, similar to (5), we have

$$\begin{aligned}
v_{j,3}^{(1)} &= v_{j,3}^{(0)} + \frac{\alpha}{n}a_j \sum_{i \in \mathcal{I}_{-\mu_1}} y_i l_i^{(0)} z_{i,3} + \frac{\alpha}{n}a_j \sum_{i \in \mathcal{I}_{[1,1,1],\mu_2} \cup \mathcal{I}_{[1,1,1],-\mu_2}} y_i l_i^{(0)} z_{i,3} \\
&\geq v_{\text{init}} - \frac{2\alpha}{n\sqrt{m}} \sum_{i \in \mathcal{I}_{-\mu_1}} l_i^{(0)} - O(\frac{\alpha}{\sqrt{m}} \max_i l_i^{(0)} \varepsilon^2 \sqrt{\frac{p}{n}} \log n) \\
&\geq v_{\text{init}} - \frac{2.1\alpha|\mathcal{I}_{-\mu_1}|}{n\sqrt{m}} \exp(v_{\text{init}} \log n) \geq v_{\text{init}} - \frac{2.1\alpha}{4\sqrt{m}}(1 + \frac{4}{\log n})(1 + 2v_{\text{init}} \log n) \\
&\geq v_{\text{init}} - \frac{3\alpha}{4\sqrt{m}} \geq \frac{v_{\text{init}}}{4},
\end{aligned} \tag{13}$$

where the last inequality comes from Assumption (A4). Then $\langle v_j^{(1)}, z\rangle < 0$ also hold for $a_j < 0$ following the same analysis. Thus we have

$$a_{j_1}\phi(\langle v_{j_1}^{(1)}, z\rangle) + a_{j_2}\phi(\langle v_{j_2}^{(1)}, z\rangle) = 0 \tag{14}$$

when $v_{j_1}^{(0)} = v_{j_2}^{(0)} = v_{\text{init}}[1, 1, 1]$ and $x_{\text{signal}} = +\mu_1$.

If $x_{\text{signal}} \in \{\pm\mu_2\}$, combining (5) and (6), we have

$$\begin{aligned}
|\langle v_j^{(1)}, z\rangle| &\leq |\langle v_j^{(0)}, \bar{z}\rangle| + |\langle v_j^{(0)}, z - \bar{z}\rangle| + |\langle v_j^{(1)} - v_j^{(0)}, \bar{z}\rangle| + |\langle v_j^{(1)} - v_j^{(0)}, z - \bar{z}\rangle| \\
&\leq 0 + v_{\text{init}}\varepsilon^2\sqrt{\frac{p}{n}}\log n + 0 + C\frac{\alpha}{\sqrt{m}}\varepsilon^2\sqrt{\frac{p}{n}}\log n \leq 2v_{\text{init}}\varepsilon^2\sqrt{\frac{p}{n}}\log n,
\end{aligned}$$

where the last inequality uses Assumption (A3) and (A4). Thus

$$|a_j\langle v_j^{(1)}, z\rangle| \leq 2v_{\text{init}}\varepsilon^2\sqrt{\frac{p}{nm}}\log n \leq \frac{2v_{\text{init}}}{\sqrt{m}\log n}. \tag{15}$$

Note that neurons initialized with $v_{\text{init}}[i, i, k], i, k \in \{\pm 1\}$ share very similar dynamics and following the same procedure, specifically, if $k = +1$ (resp. $-1$), the neurons align well with $-\mu_1$ (resp. $+\mu_1$). Additionally, the neurons do not align well with $\pm\mu_2$ for both $i = +1$ and $i = -1$. For brevity, we omit the analysis for $v_j^{(0)} = v_{\text{init}}[i, i, k], i, k \in \{\pm 1\}\setminus\{v_{\text{init}}[1, 1, 1]\}$.

Next we analyze the one-step update of neuron $v_j$ with initialization $v_{\text{init}}[1, -1, 1]$.

**(2)** If $v_j^{(0)} = v_{\text{init}}[1, -1, 1]$, then according to Lemma A.1, we have

$$\mathcal{I}_{[1,-1,1],+\mu_1} = \varnothing; \quad \mathcal{I}_{[1,-1,1],-\mu_1} = \mathcal{I}_{-\mu_1}; \quad \mathcal{I}_{[1,-1,1],\mu_2} = \mathcal{I}_{+\mu_2}; \quad \mathcal{I}_{[1,-1,1],-\mu_2} = \varnothing. \tag{16}$$

Similar to (5), we have

$$
\begin{aligned}
\left| v_{j,3}^{(1)} - \left( v_{j,3}^{(0)} + \frac{\alpha}{2} a_j \right) \right| &= \left| \frac{\alpha}{n} a_j \sum_{i=1}^{n} y_i l_i^{(0)} \phi'(\langle v_j^{(0)}, z_i \rangle) z_{i,3} - \frac{\alpha}{2} a_j \right| \\
&= \left| \frac{\alpha}{n} a_j \left[ \sum_{i \in \mathcal{I}_{-\mu_1}} y_i l_i^{(0)} z_{i,3} + \sum_{i \in \mathcal{I}_{+\mu_2}} y_i l_i^{(0)} z_{i,3} \right] - \frac{\alpha}{2} a_j \right| \\
&= \left| \frac{\alpha}{n\sqrt{m}} \left[ \sum_{i \in \mathcal{I}_{-\mu_1}} l_i^{(0)} z_{i,3} - \sum_{i \in \mathcal{I}_{+\mu_2}} l_i^{(0)} z_{i,3} \right] - \frac{\alpha}{2\sqrt{m}} \right| \\
&= \left| \frac{\alpha}{n\sqrt{m}} \left[ \sum_{i \in \mathcal{I}_{-\mu_1}} l_i^{(0)} \bar{z}_{i,3} + \sum_{i \in \mathcal{I}_{-\mu_1}} l_i^{(0)} (z_{i,3} - \bar{z}_{i,3}) - \sum_{i \in \mathcal{I}_{+\mu_2}} l_i^{(0)} (z_{i,3} - \bar{z}_{i,3}) \right] - \frac{\alpha}{2\sqrt{m}} \right| \\
&\leq \frac{2\alpha}{n\sqrt{m}} \left| \frac{n}{4} - n_{-\mu_1} \exp(-v_{\text{init}} \log n) \right| + \frac{\alpha}{n\sqrt{m}} \left( n_{-\mu_1} + n_{+\mu_2} \right) \exp(v_{\text{init}} \log n) \varepsilon^2 \sqrt{\frac{p}{n}} \log n \\
&= O\left( \frac{\alpha}{\sqrt{m}} \left( \varepsilon^2 \sqrt{\frac{p}{n}} + v_{\text{init}} \right) \log n \right) = O\left( \frac{\alpha}{\sqrt{m \log n}} \right),
\end{aligned}
\tag{17}
$$

where the first equation comes from the GD update; the second equation uses (16); the third equation uses $|a_j| = 1/\sqrt{m}$; the fourth equation uses $\bar{z}_{i,3} = 0$ for $i \in \mathcal{I}_{+\mu_2}$; the first inequality uses $\bar{z}_{i,3} = 2, i \in \mathcal{I}_{-\mu_1}$, (2) and the definition of $\mathcal{G}$; the fifth equation uses $|n_\mu - n/4| \leq n/\log n$ and $|\exp(-v_{\text{init}} \log n) - 1| \leq 2 v_{\text{init}} \log n \leq 2/\sqrt{\log n}$ by Assumption (A3); and the last equation uses Assumption (A1) and (A3). Further for the first entry of $v_j$, we have

$$
\begin{aligned}
\left| v_{j,1}^{(1)} - \left( v_{j,1}^{(0)} - \frac{\alpha}{2} a_j \right) \right| &= \left| \frac{\alpha}{n} a_j \sum_{i=1}^{n} y_i l_i^{(0)} \phi'(\langle v_j^{(0)}, z_i \rangle) z_{i,1} + \frac{\alpha}{2} a_j \right| \\
&= \left| \frac{\alpha}{n} a_j \left[ \sum_{i \in \mathcal{I}_{-\mu_1}} y_i l_i^{(0)} z_{i,1} + \sum_{i \in \mathcal{I}_{+\mu_2}} y_i l_i^{(0)} z_{i,1} \right] + \frac{\alpha}{2} a_j \right| \\
&= \left| \frac{\alpha}{n\sqrt{m}} \left[ \sum_{i \in \mathcal{I}_{-\mu_1}} l_i^{(0)} z_{i,1} - \sum_{i \in \mathcal{I}_{+\mu_2}} l_i^{(0)} z_{i,1} \right] + \frac{\alpha}{2\sqrt{m}} \right| \\
&= \left| \frac{\alpha}{n\sqrt{m}} \left[ - \sum_{i \in \mathcal{I}_{+\mu_2}} l_i^{(0)} \bar{z}_{i,1} + \sum_{i \in \mathcal{I}_{-\mu_1}} l_i^{(0)} (z_{i,1} - \bar{z}_{i,1}) - \sum_{i \in \mathcal{I}_{+\mu_2}} l_i^{(0)} (z_{i,1} - \bar{z}_{i,1}) \right] + \frac{\alpha}{2\sqrt{m}} \right| \\
&\leq \frac{2\alpha}{n\sqrt{m}} \left| 1 - n_{+\mu_2} \exp(-v_{\text{init}} \log n) \right| + \frac{\alpha}{n\sqrt{m}} \left( n_{-\mu_1} + n_{+\mu_2} \right) \exp(v_{\text{init}} \log n) \varepsilon^2 \sqrt{\frac{p}{n}} \log n \\
&= O\left( \frac{\alpha}{\sqrt{m}} \left( \varepsilon^2 \sqrt{\frac{p}{n}} + v_{\text{init}} \right) \log n \right) = O\left( \frac{\alpha}{\sqrt{m \log n}} \right),
\end{aligned}
\tag{18}
$$

where the inequality uses $\bar{z}_{i,1} = 2$ for $i \in \mathcal{I}_{+\mu_2}$. And for the second entry of $v_j$, it follows similarly that

$$
\left| v_{j,2}^{(1)} - \left( v_{j,2}^{(0)} + \frac{\alpha}{2} a_j \right) \right| = \left| \frac{\alpha}{n\sqrt{m}} \left[ \sum_{i \in \mathcal{I}_{-\mu_1}} l_i^{(0)} z_{i,2} - \sum_{i \in \mathcal{I}_{+\mu_2}} l_i^{(0)} z_{i,2} \right] - \frac{\alpha}{2\sqrt{m}} \right| = O\left( \frac{\alpha}{\sqrt{m \log n}} \right).
\tag{19}
$$

Unifying (17), (17) and (18), we obtain

$$
\left| v_{j,l}^{(1)} - \left( v_{j,l}^{(0)} + \frac{\alpha}{2} a_j \operatorname{sgn}(v_{j,l}^{(0)}) \xi_l \right) \right| = O\left( \frac{\alpha}{\sqrt{m \log n}} \right)
\tag{20}
$$

for $l = 1, 2, 3$. Here $\{\xi_l\}$ are defined as $\xi_l = -1, l = 1, 2$ and $\xi_3 = 1$.

For a datapoint $(x, y) \sim P$ with $z = [z_1, z_2, z_3]^\top = U^\top x$. We condition on the event

$$
\|z - \bar{z}\| \leq \varepsilon^2 \sqrt{\frac{p}{n}} \log n.
$$

If $x_{\text{signal}} = -\mu_1$: for each pair $j_1, j_2$ with $v_{j_1}^{(0)} = v_{j_2}^{(0)} = v_{\text{init}}[1, -1, 1]$ and $a_{j_1} > 0, a_{j_2} < 0$, we have $\langle v_{j_l}^{(1)}, z \rangle > 0, l = 1, 2$, and

$$
\| v_{j_1}^{(1)} - v_{j_2}^{(1)} - \frac{\alpha}{\sqrt{m}} [-1, 1, 1]^\top \| = \| (v_{j_1}^{(1)} - v_{j_1}^{(0)}) - (v_{j_2}^{(1)} - v_{j_2}^{(0)}) - \frac{\alpha}{\sqrt{m}} [-1, 1, 1]^\top \| = O\left( \frac{\alpha}{\sqrt{m \log n}} \right)
$$

by (20). It follows that

$$
\begin{aligned}
a_{j_1}\phi(\langle v_{j_1}^{(1)}, z\rangle) + a_{j_2}\phi(\langle v_{j_2}^{(1)}, z\rangle) &= \frac{1}{\sqrt{m}}\langle v_{j_1}^{(1)} - v_{j_2}^{(1)}, z\rangle \\
&= \frac{1}{\sqrt{m}}\Big(\langle \frac{\alpha}{\sqrt{m}}[-1,1,1], \bar{z}\rangle + \langle v_{j_1}^{(1)} - v_{j_2}^{(1)} - \frac{\alpha}{\sqrt{m}}[-1,1,1], \bar{z}\rangle + \langle v_{j_1}^{(1)} - v_{j_2}^{(1)}, z - \bar{z}\rangle\Big) \\
&\geq \frac{1}{\sqrt{m}}\Big(\frac{2\alpha}{\sqrt{m}} - O(\frac{\alpha}{\sqrt{m}\log n}) - O(\frac{\alpha}{\sqrt{m}\log n}\varepsilon^2\sqrt{\frac{p}{n}}\log n)\Big) \geq \frac{\alpha}{m}.
\end{aligned}
\tag{21}
$$

If $x_{\text{signal}} = +\mu_1$: we have $\langle v_{j_l}^{(1)}, z\rangle < 0, l = 1,2$, thus

$$
a_{j_1}\phi(\langle v_{j_1}^{(1)}, z\rangle) = a_{j_2}\phi(\langle v_{j_2}^{(1)}, z\rangle) = 0
$$

If $x_{\text{signal}} = +\mu_2$: we have $\langle v_{j_l}^{(0)}, z\rangle > 0, l = 1,2$. Applying (20) and Assumption (A4), we have $\langle v_{j_l}^{(1)}, z\rangle > 0, l = 1,2$. It follows that

$$
\begin{aligned}
a_{j_1}\phi(\langle v_{j_1}^{(1)}, z\rangle) + a_{j_2}\phi(\langle v_{j_2}^{(1)}, z\rangle) &= \frac{1}{\sqrt{m}}\langle v_{j_1}^{(1)} - v_{j_2}^{(1)}, z\rangle \\
&= \frac{1}{\sqrt{m}}\Big(\langle \frac{\alpha}{\sqrt{m}}[-1,1,1], \bar{z}\rangle + \langle v_{j_1}^{(1)} - v_{j_2}^{(1)} - \frac{\alpha}{\sqrt{m}}[-1,1,1], \bar{z}\rangle + \langle v_{j_1}^{(1)} - v_{j_2}^{(1)}, z - \bar{z}\rangle\Big) \\
&\leq \Big(-\frac{2\alpha}{\sqrt{m}} + O(\frac{\alpha}{\sqrt{m}\log n}) + O(\frac{\alpha}{\sqrt{m}\log n}\varepsilon^2\sqrt{\frac{p}{n}}\log n)\Big) \leq -\frac{\alpha}{m}
\end{aligned}
\tag{22}
$$

for sufficiently large $n$. Here the last inequality uses Assumption (A1).

If $x_{\text{signal}} = -\mu_2$: we have $\langle v_{j_l}^{(0)}, z\rangle < 0, l = 1,2$. Applying (20) and Assumption (A4), we have $\langle v_{j_l}^{(1)}, z\rangle < 0, l = 1,2$. It follows that

$$
a_{j_1}\phi(\langle v_{j_1}^{(1)}, z\rangle) + a_{j_2}\phi(\langle v_{j_2}^{(1)}, z\rangle) = 0.
\tag{23}
$$

In conclusion, for datapoint $(x,y)$ with $x_{\text{signal}} = -\mu_1$, conditioning on (7), the output of $f^{(1)}$ is

$$
\begin{aligned}
f^{(1)}(x) = \sum_{j=1}^{m} a_j\phi(\langle v_j^{(1)}, z\rangle) &= \sum_{e\in\text{Uniform}(\{\pm1\}^3)}\sum_{j\in\mathcal{J}_e} a_j\phi(\langle v_j^{(1)}, z\rangle) \\
&= \sum_{e:e_3=1}\Big[\sum_{j\in\mathcal{J}_{\text{Pos},e}} a_j\phi(\langle v_j^{(1)}, z\rangle) - \sum_{j\in\mathcal{J}_{\text{Neg},e}} a_j\phi(\langle v_j^{(1)}, z\rangle)\Big] \\
&\geq \sum_{e:e_3=1}\Big[\min\{|\mathcal{J}_{\text{Pos},e}|, |\mathcal{J}_{\text{Neg},e}|\}\frac{2\alpha}{5m} - \frac{4\sqrt{m}}{\log n}(v_{\text{init}} + \frac{\alpha}{\sqrt{m}})\Big] \\
&\geq \sum_{e:e_3=1}\Big[\frac{\alpha}{40} - \frac{4\sqrt{m}}{\log n}(C\frac{\alpha}{\sqrt{m}} + \frac{\alpha}{\sqrt{m}})\Big] > 0
\end{aligned}
\tag{24}
$$

for sufficiently large $n$. Here the first inequality uses (12) and (21), the property that $\big||\mathcal{J}_{\text{Pos},e}| - |\mathcal{J}_{\text{Neg},e}|\big| \leq 2m/\log n$ from (B3) in Lemma A.4 and the property that $\phi(\langle v_j^{(1)}, z\rangle) \leq 2(v_{\text{init}} + \alpha/\sqrt{m})$; the second inequality uses (B3) and Assumption (A4). Similarly, we have that for datapoint $(x,y)$ with $x_{\text{signal}} = +\mu_1$, conditioning on (7), the output of $f^{(1)}$ is

$$
f^{(1)}(x) = \sum_{e:e_3=-1}\Big[\sum_{j\in\mathcal{J}_{\text{Pos},e}} a_j\phi(\langle v_j^{(1)}, z\rangle) - \sum_{j\in\mathcal{J}_{\text{Neg},e}} a_j\phi(\langle v_j^{(1)}, z\rangle)\Big] > 0.
\tag{25}
$$

For datapoint $(x, y)$ with $x_{\text{signal}} = +\mu_2$, conditioning on (7), the output of $f^{(1)}$ is

$$
\begin{aligned}
f^{(1)}(x) &= \sum_{j=1}^{m} a_j \phi(\langle v_j^{(1)}, z \rangle) = \sum_{e \in \text{Uniform}(\{\pm 1\}^3)} \sum_{j \in \mathcal{J}_e} a_j \phi(\langle v_j^{(1)}, z \rangle) \\
&= \Big( \sum_{e:[e_1,e_2]=[1,-1]} + \sum_{e:e_1=e_2} \Big) \Big[ \sum_{j \in \mathcal{J}_{\text{Pos},e}} a_j \phi(\langle v_j^{(1)}, z \rangle) - \sum_{j \in \mathcal{J}_{\text{Neg},e}} a_j \phi(\langle v_j^{(1)}, z \rangle) \Big] \\
&\leq \sum_{e:[e_1,e_2]=[1,-1]} \Big[ -\min\{|\mathcal{J}_{\text{Pos},e}|, |\mathcal{J}_{\text{Neg},e}|\} \frac{\alpha}{m} + \frac{4\sqrt{m}}{\log n}(v_{\text{init}} + \frac{\alpha}{\sqrt{m}}) \Big] + \sum_{e:e_1=e_2} \frac{2 v_{\text{init}} |\mathcal{J}_{\text{Neg},e}|}{\sqrt{m} \log n} \\
&\leq 2\Big( -\frac{\alpha}{16} + \frac{5\sqrt{m}}{\log n}(v_{\text{init}} + \frac{\alpha}{\sqrt{m}}) \Big) + \frac{v_{\text{init}} \sqrt{m}}{\sqrt{\log n}} \leq -\frac{\alpha}{8} + \frac{10(C+1)\alpha}{\log n} + \frac{C\alpha}{\sqrt{\log n}} < 0,
\end{aligned}
\tag{26}
$$

where the first inequality uses (15), (22), (B3) and the property that $\phi(\langle v_j^{(1)}, z \rangle) \leq 2(v_{\text{init}} + \alpha/\sqrt{m})$; the second inequality uses (B3); and the third inequality uses Assumption (A4). Similarly, we have that for datapoint $(x, y)$ with $x_{\text{signal}} = -\mu_2$, conditioning on (7), $f^{(1)}(x) < 0$, which combined with (24), (25) and (26), yields that

$$
\text{sgn}(f^{(1)}(x)) = y
$$

for any $(x, y) \sim P, z = U^\top x$ satisfying

$$
\|z - \bar{z}\| \leq \varepsilon^2 \sqrt{\frac{p}{n}} \log n.
$$

According to the definition of $\mathcal{G}_{\text{data}}$, all $(x_i, y_i)$ satisfy this condition. Thus conditioning on the event $\mathcal{G}$, the model $f^{(1)}$ can correctly classify all training data points. And applying the law of total probability, we obtain that the test error is bounded by:

$$
\begin{aligned}
\mathbb{P}_{(x,y) \sim P}(y \neq \text{sgn}(f^{(1)}(x))) &\leq \mathbb{P}_{(x,y) \sim P}\Big(y \neq \text{sgn}(f^{(1)}(x)) \mid \|z - \bar{z}\| \leq \varepsilon^2 \sqrt{\frac{p}{n}} \log n \Big) \\
&\quad + \mathbb{P}_{(x,y) \sim P}\Big(\|z - \bar{z}\| \leq \varepsilon^2 \sqrt{\frac{p}{n}} \log n \Big) \\
&= \mathbb{P}_{(x,y) \sim P}\Big(\|z - \bar{z}\| \leq \varepsilon^2 \sqrt{\frac{p}{n}} \log n \Big) \leq \exp(-\Omega(\log^2 n)),
\end{aligned}
$$

where the last inequality uses Lemma A.3. $\qquad \square$

**Lemma A.1.** *Suppose that Assumption (A2) holds. With probability at least $1 - O(\frac{1}{n^2})$, the following conditions hold:*

$$
\begin{aligned}
\mathcal{I}_{[i,j,-1],+\mu_1} &= \mathcal{I}_{+\mu_1}; \quad \mathcal{I}_{[i,j,-1],-\mu_1} = \varnothing, \quad i, j \in \{\pm 1\}; \\
\mathcal{I}_{[i,j,+1],+\mu_1} &= \varnothing; \quad \mathcal{I}_{[i,j,+1],-\mu_1} = \mathcal{I}_{-\mu_1}, \quad i, j \in \{\pm 1\}; \\
\mathcal{I}_{[+1,-1,k],+\mu_2} &= \mathcal{I}_{+\mu_2}; \quad \mathcal{I}_{[+1,-1,k],-\mu_2} = \varnothing, \quad k \in \{\pm 1\};
\end{aligned}
\tag{27}
$$

$$
\begin{aligned}
\mathcal{I}_{[-1,+1,k],+\mu_2} &= \varnothing; \quad \mathcal{I}_{[-1,+1,k],-\mu_2} = \mathcal{I}_{-\mu_2}, \quad k \in \{\pm 1\}; \\
\Big| |\mathcal{I}_{[i,i,k],\mu}| - \frac{n_\mu}{2} \Big| &\leq \sqrt{n \log n}, \quad i, k \in \{\pm 1\}, \mu \in \{\pm \mu_2\}.
\end{aligned}
\tag{28}
$$

*Proof.* For simplicity, we denote $\mathbb{P}(\cdot \mid \{(x_i, y_i)\}_{i=1}^n \in \mathcal{G}_{\text{data}})$ as $\mathbb{P}(\cdot)$ in the proof below.

For $v_j^{(0)} = [v_{\text{init}}, v_{\text{init}}, v_{\text{init}}]$, we first show that for $\{(x_i, y_i)\}_{i=1}^n \in \mathcal{G}_{\text{data}}$,

$$
\langle v_j^{(0)}, z_i \rangle > 0, \quad \forall i \in \mathcal{I}_{-\mu_1}.
$$

According to the definition of $\mathcal{G}_{\text{data}}$, $\|z_i - \bar{z}_i\| \leq \varepsilon^2 \sqrt{p/n} \log n$ for all $i \in \mathcal{I}_{-\mu_1}$. Thus

$$
\langle v_j^{(0)}, z_i \rangle = \langle v_j^{(0)}, \bar{z}_i \rangle + \langle v_j^{(0)}, z_i - \bar{z}_i \rangle \geq v_{\text{init}}(2 - \|z_i - \bar{z}_i\|) > 0,
$$

where the first inequality uses $\bar{z}_i = [0, 0, 2]$ when $i \in \mathcal{I}_{-\mu_1}$ and the second inequality uses Assumption (A1). Similarly, we have

$$\langle v_j^{(0)}, z_i \rangle < 0, \quad \forall i \in \mathcal{I}_{+\mu_1}.$$

Thus conditioning on $\{(x_i, y_i)\}_{i=1}^n \in \mathcal{G}_{\text{data}}$, we have

$$\mathcal{I}_{[1,1,1],-\mu_1} = \mathcal{I}_{-\mu_1}; \mathcal{I}_{[1,1,1],+\mu_1} = \varnothing.$$

For $i \in \mathcal{I}_{\pm\mu_2}$, recall that $x_i = [x_{i,\text{signal}}^\top, x_{i,\text{noise}}^\top]^\top$ with $x_{i,\text{signal}} = [x_{i,1}, x_{i,2}]^\top$ and $x_{i,\text{noise}} = [x_{i,3}, \cdots, x_{i,p}]^\top$. We have

$$\langle v_j^{(0)}, z_i \rangle = v_{\text{init}} \sum_{l=1}^3 z_{i,l} = v_{\text{init}} (\sum_{l=1}^3 u_l)^\top x_i = v_{\text{init}} [-\mu_1^\top, (\sum_{l=1}^3 \delta_l)^\top] x_i = v_{\text{init}} (\sum_{l=1}^3 \delta_l)^\top x_{i,\text{noise}}.$$

It follows that

$$\mathbb{P}(\langle v_j^{(0)}, z_i \rangle > 0) = \frac{1}{2}.$$

Applying Hoeffding's inequality, we obtain

$$\mathbb{P}\left(\left||\mathcal{I}_{[1,1,1],+\mu_2}| - \frac{|\mathcal{I}_{+\mu_2}|}{2}\right| > t\right) \leq 2\exp(-\frac{2t^2}{n}).$$

Similarly we have

$$\mathbb{P}\left(\left||\mathcal{I}_{[1,1,1],-\mu_2}| - \frac{|\mathcal{I}_{-\mu_2}|}{2}\right| > t\right) \leq 2\exp(-\frac{2t^2}{n}).$$

Let $t = \sqrt{n \log n}$. We have

$$\left||\mathcal{I}_{[1,1,1],\mu}| - \frac{n_\mu}{2}\right| \leq \sqrt{n \log n}, \quad \mu \in \{\pm\mu_2\}$$

with probability at least $1 - 4/n^2$. Following similar discussion, we have that

$$\mathcal{I}_{[i,j,-1],+\mu_1} = \mathcal{I}_{+\mu_1}; \quad \mathcal{I}_{[i,j,-1],-\mu_1} = \varnothing, \quad i,j \in \{\pm 1\};$$

$$\mathcal{I}_{[i,j,+1],+\mu_1} = \varnothing; \quad \mathcal{I}_{[i,j,+1],-\mu_1} = \mathcal{I}_{-\mu_1}, \quad i,j \in \{\pm 1\};$$

$$\mathcal{I}_{[+1,-1,k],+\mu_2} = \mathcal{I}_{+\mu_2}; \quad \mathcal{I}_{[+1,-1,k],-\mu_2} = \varnothing, \quad k \in \{\pm 1\};$$

$$\mathcal{I}_{[-1,+1,k],+\mu_2} = \varnothing; \quad \mathcal{I}_{[-1,+1,k],-\mu_2} = \mathcal{I}_{-\mu_2}, \quad k \in \{\pm 1\}$$

hold with probability 1 given $\{(x_i, y_i)\}_{i=1}^n \in \mathcal{G}_{\text{data}}$. And

$$\left||\mathcal{I}_{[i,i,k],\mu}| - \frac{n_\mu}{2}\right| \leq \sqrt{n \log n}, \quad i,k \in \{\pm 1\}, \mu \in \{\pm\mu_2\}$$

hold with probability at least $1 - 16/n^2$. In total, the conditions above hold with probability at least $1 - \exp(-\Omega(\log^2 n)) - O(\frac{1}{n^2}) = 1 - O(\frac{1}{n^2})$.

$\square$

**Lemma A.2.** *Suppose that Assumption (A2) holds. Let the training data $\{x_i, y_i\}_{i=1}^n$ for model $f_L$ be sampled i.i.d from $P$. With probability at least $1 - \exp(-\Omega(\log^2 n))$, we have*

$$\|z_i - \bar{z}_i\| \leq \varepsilon^2 \sqrt{\frac{p}{n}} \log n, \quad \text{for all } i \in [n]. \tag{29}$$

*Proof.* Applying Lemma A.3, we obtain

$$\mathbb{P}(\|z_i - \bar{z}_i\| \leq \varepsilon^2 \sqrt{\frac{p}{n}} \log n, \forall i \in [n]) \geq 1 - \sum_{i=1}^n \mathbb{P}(\|z_i - \bar{z}_i\| > \varepsilon^2 \sqrt{\frac{p}{n}} \log n)$$

$$\geq 1 - n\exp(-\Omega(\log^2 n)) = 1 - \exp(-\Omega(\log^2 n)).$$

$\square$

**Lemma A.3.** *Suppose that Assumption (A2) holds. For $x = [x_1, x_2, \cdots, x_p] \sim P$ with $[x_1, x_2]^\top = \mu, \mu \in \{\pm\mu_1, \pm\mu_2\}$, we have*

$$\mathbb{P}(\|U^\top x - \nu\|_{\max} > \varepsilon^2 \sqrt{\frac{p}{n}} \log n) \le \exp(-\Omega(\log^2 n)), \tag{30}$$

*where $\nu = [\mu_2, -\mu_2, -\mu_1]^\top \mu \in \mathbb{R}^3$.*

*Proof.* We start our analysis with $[x_1, x_2]^\top = \mu_1$. Note that

$$u_1^\top x = \sum_{i=1}^{p-2} \delta_{1,i} x_{i+2}$$

is a summation of independent bounded random variables with zero mean. By Hoeffding's inequality, we have

$$\mathbb{P}(|u_1^\top x| \ge t) \le 2\exp\left(-\frac{t^2}{2\sum_{i=1}^{p-2} \delta_{1,i}^2 \varepsilon^2}\right) \le 2\exp\left(-\frac{t^2}{2\|\delta\|_{\mathrm{F}}^2 \varepsilon^2}\right).$$

Similarly, the concentration for $u_2^\top x$ and $u_3^\top x + 2$ are as follows:

$$\mathbb{P}(|u_2^\top x| \ge t) \le 2\exp\left(-\frac{t^2}{2\|\delta\|_{\mathrm{F}}^2 \varepsilon^2}\right);$$

$$\mathbb{P}(|u_3^\top x + 2| \ge t) \le 2\exp\left(-\frac{t^2}{2\|\delta\|_{\mathrm{F}}^2 \varepsilon^2}\right).$$

Combining these inequalities yields

$$\mathbb{P}(\|U^\top x - \nu_1\|_{\max} > t) \le 6\exp\left(-\frac{t^2}{2\|\delta\|_{\mathrm{F}}^2 \varepsilon^2}\right) \le 6\exp\left(-\frac{nt^2}{2C^2\varepsilon^4 p}\right), \tag{31}$$

where the last inequality uses Assumption (A2). The proof concludes by letting $t = \varepsilon^2 \sqrt{p/n} \log n$. The analysis for other values of $[x_1, x_2]^\top$ follows similarly. $\square$

**Lemma A.4.** *Suppose that Assumption (A5) holds. Then the following conditions hold with probability at least $1 - O(1/n^4)$:*

*(B1)* $\max_{k \in \{\mathrm{Pos}, \mathrm{Neg}\}} \left||\mathcal{J}_k| - \frac{m}{2}\right| \le \frac{m}{\log n}$.

*(B2)* $\max_{e \in \mathit{Uniform}(\{\pm 1\}^3)} \left|\mathcal{J}_e - \frac{m}{8}\right| \le \frac{m}{\log n}$

*(B3)* $\max_{k \in \{\mathrm{Pos}, \mathrm{Neg}\}, e \in \mathit{Uniform}(\{\pm 1\}^3)} \left|\mathcal{J}_{k,e} - \frac{m}{16}\right| \le \frac{m}{\log n}$.

*(B4)* $\max_{\mu \in \{\pm\mu_1, \pm\mu_2\}} \left|n_\mu - \frac{n}{4}\right| \le \frac{n}{\log n}$.

*Proof.* Note that $|\mathcal{J}_{\mathrm{Pos}}| \sim \mathrm{Bin}(m, 1/2)$. Applying Hoeffding's inequality, we have

$$\mathbb{P}\left(\left||\mathcal{J}_{\mathrm{Pos}}| - \frac{m}{2}\right| \le \frac{m}{\log n}\right) \le 2\exp\left(-\frac{2m}{\log^2 n}\right) \le \frac{2}{n^4},$$

where the last inequality comes from Assumption (A5). And similarly

$$\mathbb{P}\left(\left||\mathcal{J}_{\mathrm{Neg}}| - \frac{m}{2}\right| \le 2\exp\left(-\frac{2m}{\log^2 n}\right) \le \frac{2}{n^4},$$

which completes the proof of (B1). Note that $|n_\mu| \sim \mathrm{Bin}(n, 1/4)$. Applying Hoeffding's inequality, we have

$$\mathbb{P}\left(\left||n_\mu| - \frac{n}{4}\right| \le \frac{n}{\log n}\right) \le 2\exp\left(-\frac{2n}{\log^2 n}\right) = O(\frac{1}{n^4}), \quad \forall \mu \in \{\pm\mu_1, \pm\mu_2\}.$$

(B2)-(B3) can be proved following the same procedure. We omit the proof here. $\square$

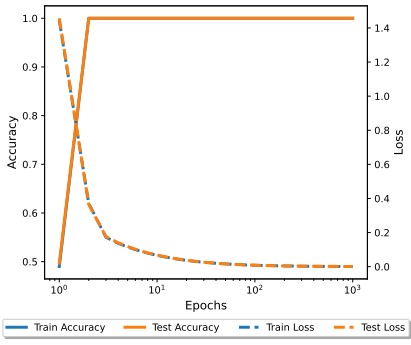

Figure 9: Training dynamics of the model $f_L$ discussed in Section 3.3.

## A.2 Additional Experiments

## A.3 Experimental details

All experiments in the paper can be run on a single NVIDIA A100 GPU. The loss function for modular arithmetic tasks is cross-entropy loss and for $(40, 3)$-parity task is logistic loss. All models used in the paper, unless stated otherwise, set $d_{mlp} = 4d_{embed}$, where $d_{mlp}$ is the MLP dimension and $d_{embed}$ is the embedding dimension. All FNN models used are in the paper are homogeneous and do not have bias terms. Code is available at https://github.com/zhiweixx/groktransfer.

### A.3.1 Experiments in Section 1 and 2.1

In Figure 1b, we use a two-layer FNN with trainable embedding layer as the weak model. We choose $(d_{embed}, \text{width}) = (4, 16)$ for the weak model. The target model is a three-layer FNN with trainable embedding layer. We choose $(d_{embed}, \text{width}) = (128, 512)$ for the target model. The hyperparameters $(\text{init scale}, \text{learning rate}, \text{weight decay})$ are selected by the following grid search:

$$\text{init scale: } [0.1, 0.2, \cdots, 1.5]$$
$$\text{learning rate: } [10^{-4}, 5 \times 10^{-4}, 10^{-3}, 5 \times 10^{-3}, 10^{-2}, 10^{-1}]$$
$$\text{weight decay: } [10^{-4}, 10^{-3}, 10^{-2}, 10^{-1}, 1, 2, 3, 4, 5].$$

We select the configuration that first achieves $90\%$ accuracy on the validation set. The best configuration for `GrokTransfer` is $(0.3, 0.005, 3)$. For standard training, only learning rate and weight decay are tuned. They are selected by the following grid search:

$$\text{learning rate: } [10^{-3}, 5 \times 10^{-3}, 10^{-2}, 5 \times 10^{-2}, 10^{-1}]$$
$$\text{weight decay: } [10^{-2}, 10^{-1}, 1, 2, 3, 4, 5],$$

and the optimal configuration is $(0.05, 3)$.

In Figure 2, we set the dimension of the GPT embedding to be $128$. For the Fourier embedding, we choose $k = 7$ frequencies, and let $i_j$ to be the $j$-th smallest prime number. For each type of embedding, we normalize the embedding of each integer to be $1$. The FNN used in Figure 2 is a three layer dense neural network

$$f(x) = W_3 \phi(W_2 \phi(W_1 x)),$$

where $W_1 \in \mathbb{R}^{\text{width} \times \text{embed dim}}, W_2 \in \mathbb{R}^{\text{width} \times \text{width}}, W_3 \in \mathbb{R}^{p \times \text{width}}$, width$= 512$. The hyperparameters $(\text{init scale}, \text{learning rate}, \text{weight decay})$ are selected by the following grid search:

$$\text{init scale: } [0.1, 0.2, \cdots, 1.5]$$
$$\text{learning rate: } [10^{-4}, 10^{-3}, 10^{-2}, 10^{-1}, 1]$$
$$\text{weight decay: } [10^{-4}, 10^{-3}, 10^{-2}, 10^{-1}, 1, 5, 10].$$

We select the configuration that first achieves $90\%$ accuracy on the validation set. The best configuration $(\text{init}, \text{lr}, \text{wd})$ for the four embeddings are:

One-hot: $(0.2, 0.01, 5)$;    Binary: $(0.3, 0.01, 1)$;    Fourier: $(0.5, 0.1, 0.1)$    GPT: $(1.3, 0.01, 1)$.

In Figure 3, the distance between two empirical NTK is estimated following the method in Mohamadi et al. (2023). We denote $\widehat{\Theta}_t$ as the pseudo-NTK of the model at epoch $t$, i.e.

$$\widehat{\Theta}_t(x_1, x_2) = [\nabla_\theta \sum_{i=1}^p f_\theta^{(i)}(x_1)]^\top [\nabla_\theta \sum_{i=1}^p f_\theta^{(i)}(x_2)]/p \in \mathbb{R}.$$

We estimate the distance between the empirical NTK at step $t$ and $t-1$ by $\|\widehat{\Theta}_t - \widehat{\Theta_{t-1}}\|_{\mathbf{F}}$.

### A.3.2   EXPERIMENTS IN SECTION 3

For experiments in Section 3, we let the sample size $n = 400$, feature dimension $p = 80000$, and noise level $\varepsilon = 0.05$. For Figure 4(a), the model is a two-layer neural network with width 2048. The optimizer is full-batch gradient descent with learning rate 0.1 and weight decay 0.1. In Figure 4(b), we train a small model with only three neurons. The initialization of the hidden layer follows i.i.d $N(0, 0.01)$, and the initialization of the second layer follows i.i.d $N(0, 10^{-4})$. The learning rate is 0.1 and weight decay is $0, 1$. Figure 4(c) visualizes the hidden layer of that small model after training.

Figure 5a generates $4000$ i.i.d datapoints from the distribution $P$, and visualizes $Ux$ for each $x$. Figure 5b fixes $n = 1000$ and train the weak model for $p = [4 \times 10^4, 8 \times 10^4, 16 \times 10^4,], \epsilon = [1/40, 1/80, 1/160]$. Figure 5c fixes $p = 8 \times 10^4$ and train the weak model for $n = [400, 800, 1600, 3200], \epsilon = [1/40, 1/80, 1/160]$. Figure 9 takes $v_{\text{init}} = 0.4$, learning rate 2.0 and zero weight decay.

### A.3.3   EXPERIMENTS IN SECTION 4

The attention layer used in this paper follows the same structure as that in Nanda et al. (2023). While Nanda et al. (2023) also suggested to set the precision to be `float64` to mitigate the Slingshot phenomenon (Thilak et al., 2022), a fluctuation of accuracy and loss during training process, we still use `float32` to control the computation cost.

All modular tasks set $p = 113$ and the fraction of training data being $25\%$.

For the $(40, 3)$-parity task, we set the sample size $n = 1000$.

Unless otherwise specified, we use the AdamW optimizer (Loshchilov & Hutter, 2019) for all experiments; we initialize the weights using the default PyTorch initialization scaled by a factor init scale $> 0$ to control the initial weight norm, as proposed by Liu et al. (2023).

The hyperparameters (init scale, learning rate, weight decay) are selected by the following grid search:

$$\text{init scale: } [0.05, 0.1, 0.2, 0.3, \cdots, 1.5]$$
$$\text{learning rate: } [10^{-4}, 5 \times 10^{-4}, 10^{-3}, 5 \times 10^{-3}, 10^{-2}, 10^{-1}, 0.5, 1.0]$$
$$\text{weight decay: } [10^{-4}, 10^{-3}, 10^{-2}, 10^{-1}, 1, 2, 3, 4, 5].$$

In Figure 6(a),(b), the structure of weak and target model are the same as those in Figure 1b. In Figure 6(c), the weak model is a three-layer $width = 16$ FNN, the target model is a three-layer $width = 512$ FNN with $d_{embed} = 128$. In Figure 6(a), the optimal configuration for `GrokTransfer` is $(0.3, 0.001, 1)$ and the optimal one for training from scratch is $(0.1, 0.1, 2)$. In Figure 6(b), the optimal configuration for `GrokTransfer` is $(0.3, 0.005, 3)$ and the optimal one for training from scratch is $(0.1, 0.1, 2)$. In Figure 6(b), we have the number of training samples $n = 1000$. The model trained via `GrokTransfer` uses learning rate $10^{-3}$ and weight decay $10^{-3}$; the model trained from scratch uses learning rate $10^{-2}$ and weight decay 1.

In Figure 8(a), the weak model is a two-layer width-4 FNN, and the target model is an 8-layer transformer with $d_{embed} = 512, d_{mlp} = 512, n_{head} = 4, d_{head} = 128$. The optimal configuration for target model trained via `GrokTransfer` is $(0.7, 0.001, 1)$. The optimal configuration for target model trained from scratch is $(0.4, 0.0005, 1)$. In Figure 8(b), the weak model remains the same, and the target model becomes an 2-layer transformer with $d_{embed} = 128, d_{mlp} = 128, n_{head} = 4, d_{head} = 32$. The optimal configuration for target model trained via `GrokTransfer` is $(0.6, 0.005, 0.1)$. The optimal configuration for `GrokFast` is (lr, wd) = $(0.01, 1.0)$.

### A.4 ADDITIONAL DISCUSSION

#### A.4.1 FORWARD PASS FLOPS ESTIMATION FOR MODELS IN FIGURE 8 LEFT

For the weak model, a two-layer width-4 MLP:

$$C_{forward} \approx 2 * (8 * 4 + 8 + 4 * 113 + 226) = 1436 \sim 10^3.$$

For the target model, an 8-layer transformer, following Table 1 in Kaplan et al. (2020), we have

$$N = 2d_{embed}n_{layer}(2d_{attn} + d_{mlp}) = 2*512*8*(256+512) = 6291456, C_{forward} = 2(N+8*2*128) \sim 10^7.$$

Two-layer FNN takes around 2000 epochs to generalize. Target model trained by `GrokTransfer` takes around 1000 epochs and target model trained from scratch takes around 10000 epochs. Thus, the total flops of `GrokTransfer` is around $10^{10} + 2 * 10^6$ and the total flops of training from scratch is around $10^{11}$.

#### A.4.2 ADDITIONAL EXPERIMENTS

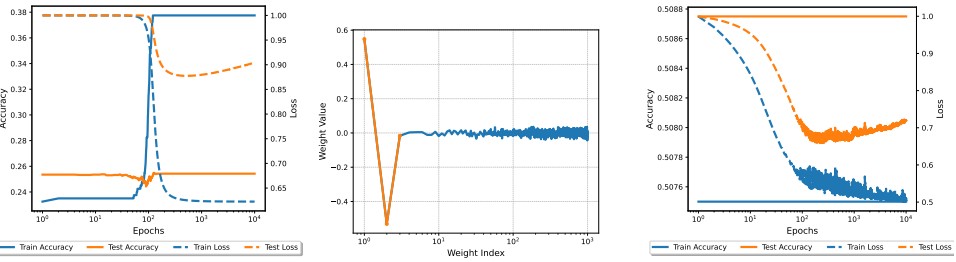

Figure 10: Left: Training dynamics of the one neuron weak model. Middle: Visualization of the neuron in the weak model. We can see it has learned the feature $[1, -1]$. Right: training dynamics of the target model with embedding transferred from the one-neuron weak model.

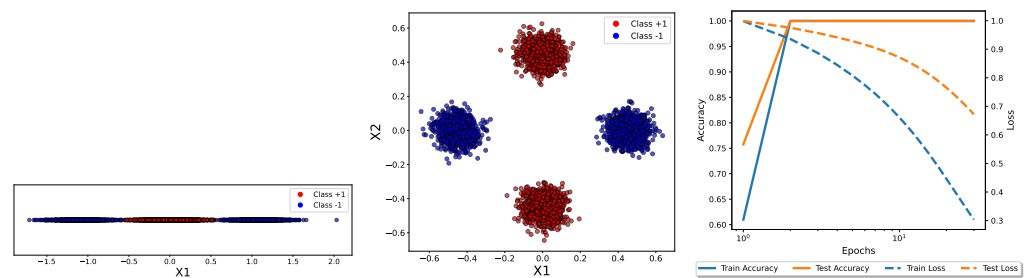

Figure 11: Left: Visualization of the distribution $P$ with the embedding from the one-neuron weak model. Middle: Visualization of the distribution $P$ with the embedding from the two-neuron weak model. Right: Training dynamics of the target model with embedding transferred from a two-neuron weak model.

When the weak model only has one neuron, the target model has the form:

$$f_L(x) = \sum_{j=1}^{m} a_j \phi(v_j \cdot u^\top x).$$

Denote $z = u^\top x$. Equivalently, we have

$$f_L(z) = \sum_{j=1}^{m} a_j \phi(v_j \cdot z).$$

Note that $\text{sign}(f_L(z_1)) = \text{sign}(f_L(z_2))$ if $\text{sign}(z_1) = \text{sign}(z_2)$. Since both positive and negative classes locate on both sides of the original point (Figure 11 Left), it is impossible for $f_L$ to correctly classify these points (Figure 10 Right).

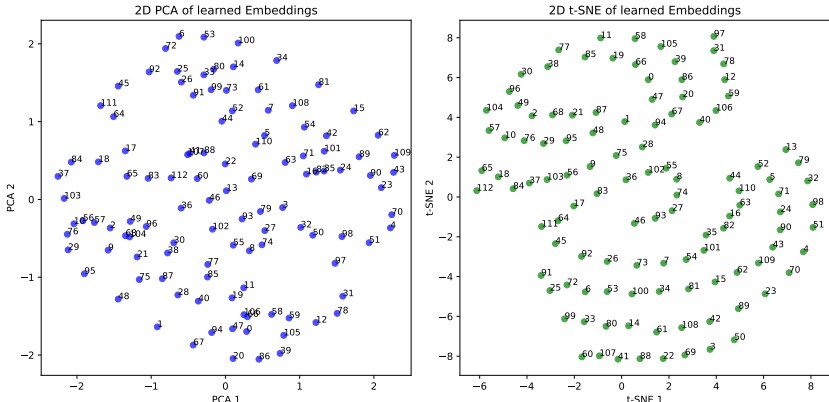

Figure 12: 2D Visualization of the 4D embedding from the weak model trained on modular addition task.

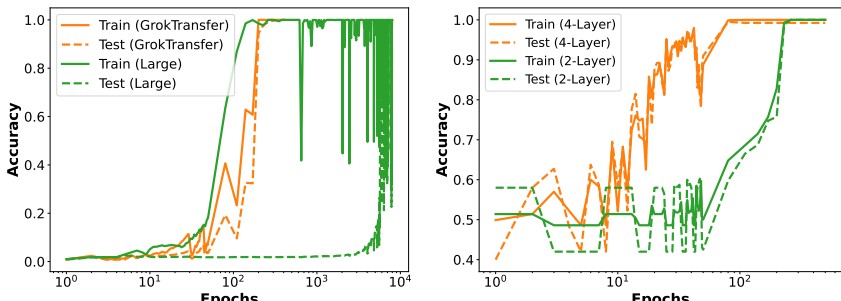

Figure 13: Left: Training dynamics of an 8-layer transformer on modular multiplication task. Right: Training dynamics of a 2-layer transformer and a 4-layer transformer trained on the $(40, 3)$-parity task. Both do not have delayed generalization.

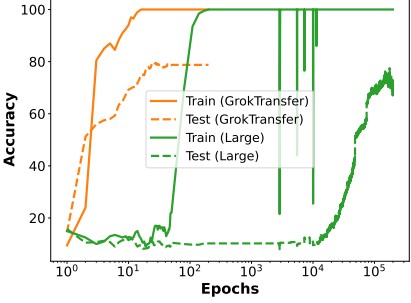

Figure 14: Training dynamics of a four-layer MLP on MNIST dataset. $200$ samples are randomly selected as the training data, and $400$ samples are randomly selected as the test data. The weak model is a two-layer MLP with $40\%$ test accuracy.

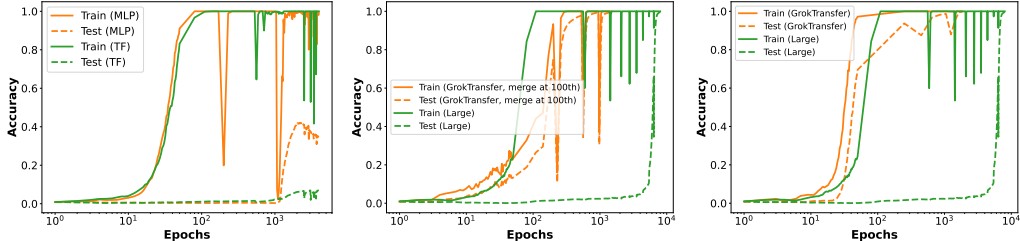

Figure 15: (a)Training dynamics of the target models used in Figure 6a and Figure 8 Right, but with low-rank embedding $A \cdot B$, both $A$ and $B$ are randomly initialized. MLP represents a three-layer MLP and TF represents a two-layer transformer. (b) Merge $A$ and $B$ into $E_T$ at 100-th epoch and keep training. (c) Set the embedding dimension of the weak model to be the same as that of the target model.

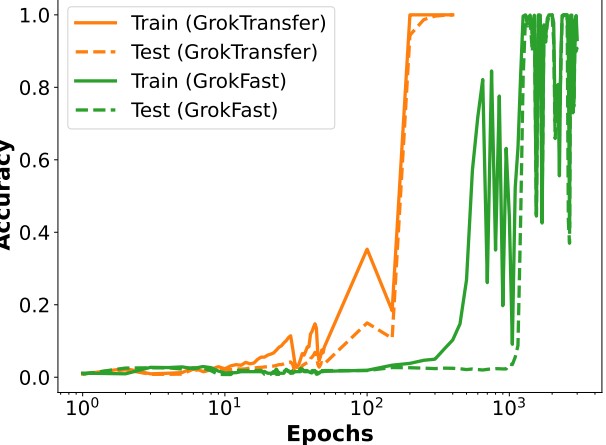

Figure 16: Dynamics of the target model (a two-layer Transformer) trained via `GrokTransfer`, and the target model trained via GrokFast on modular multiplication task.

