# OpenReview forum: "Let Me Grok for You: Accelerating Grokking via Embedding Transfer from a Weaker Model"
_ICLR.cc/2025/Conference — ICLR 2025 Poster_

### Official Review · Reviewer_s3cV · 2024-10-31

**Soundness:** 2
**Presentation:** 4
**Contribution:** 2
**Rating:** 3
**Confidence:** 4

**Summary:**

The paper introduces a method to mitigate grokking, which describes the scenario in which models overfit to the training dataset long before it begins to generalize to unseen test data. This is done by training a smaller model with a much lower embedding (number of features in first layer) dimension, which converges quicker and without grokking. The architecture of the original model is then modified by replacing the first embedding layer with the product of two matrices $E_T = AB$ where $A$ and $B$ are of much lower rank. $A$ is initialized with the embedding layer of the smaller model. This model is evaluated on several synthetic tasks -- modular addition, modular multiplication, and a (40,3)-parity task defined as learning $y = \Pi_{i \in \{1,2,3\}} x_i$ where $x \in \{\pm 1\}^{40}$. Experiments show that compared to the original model that is randomly initialized, the resulting model mitigates grokking and converges faster. Theoretical analysis is also provided for a specific case involving learning $y = x_1$  XOR $x_2$ for a 80K dimensional vector $x$.

**Strengths:**

- The method is clearly described and easily understandable and implementable.
- The combined time for the proposed method appears much faster than the baseline method of training the original target model from scratch. Table 1 suggests it is more than 5x faster. However ablations are needed to determine whether the speedup comes from architectural modifications, or from the proposed method.
- For the tasks studied, the model performs well in terms of reducing grokking and accelerating convergence (although comparisons are not apples-to-apples -- see Weakness 1)

**Weaknesses:**

- My main concern is that all the numbers reported for GrokTransfer and the target model (Large) are not directly/fairly comparable. The model used for GrokTransfer parameterizes the first layer with $E_T = AB$ where $A$ and $B$ are (very) low-rank matrices. On the other hand, the original target model uses the full $d_v \times d_T$ matrix which is significantly larger. The number of parameters present in this layer is now much larger than that of $E_T = AB$ . A fair comparison should not even consider target models with the full $d_v \times d_T$ embedding layer, but only those parametrized with $AB$ , where, for instance, $A$ can be randomly initialized instead of being initialized from the weaker model. As such, it is difficult to conclude anything regarding whether the improvement comes from weak model initialization, or simply from architectural modifications, from any of the existing results.
- This defeats the purpose of studying/mitigating grokking in the first place, where the goal is to be able to train over-parametrized models with minimal grokking. This method instead replaces the over-parametrized model with one of more manageable complexity, which ties to the next weakness
- Method also greatly reduces the expressivity of the original target model, by adding a very low-rank bottleneck to the first layer. While this works for the synthetic closed-form tasks considered in the paper, it suggests that (1) the original target model architecture is clearly ill-suited, since it converges perfectly even after introducing a very low-rank bottleneck, and (2) ability of the method to generalize to more complex tasks which require greater expressivity is limited and questionable.
- The general motivation of the method appears to be applying dimensionality reduction to pre-process inputs that have very low SNR. This opens up questions regarding how the method compares to classical techniques like LDA?
- Method introduces an additional layer of complexity in the training pipeline, due to having to train an additional model which would require its own architectural and hyperparameter sweeps.
- Minor comments on notation: Inconsistent notation switching between $d_{embed}$ and $d_W$  / $d_F$, which makes it hard to look up. Notation on line $y=x_1 x_2$ is also ambiguous, I assume the multiplication operation here is XOR. Also overloaded notation for $p$ for modulo and dimension of input in XOR task.

**Questions:**

- Do the conditions in the theoretical analysis, under which grokking is mitigated for the XOR task, have any implications on what are the causes of grokking? For instance, it possibly suggests several several causes of grokking, including the SNR, overparametrization (A5), initialization (A3), etc.
- Are there failure cases of the method? I do not see mentions of them in the limitations.

---

> ### Author Response · Authors · 2024-11-23
> **Response to Weaknesses**
>
> We thank the reviewer for their review and address their comments below. We hope that the response satisfactorily addresses the reviewer’s questions and that the reviewer will consider raising their score in light of our response.
>
> - ***My main concern is that all the numbers reported for GrokTransfer and the target model (Large) are not directly/fairly comparable.......***
> - ***This defeats the purpose of studying/mitigating grokking in the first place, ......***
> - ***Method also greatly reduces the expressivity of the original target model, by adding a very low-rank bottleneck to the first layer.......***
>
>
> We would like to clarify that the low-rank bottleneck is not a crucial aspect of GrokTransfer.
>
> Firstly, we found that $A$ and $B$ can be merged into the full matrix $E_T$ later on during training, and $E_T$ can be further trained to convergence. This has little influence on the performance, and doing this will not impose any low-rank constraint. We have included this experiment in **Figure 15(b)** (page 28), where we first train the target model with low-rank embedding for 100 epochs, merge $A, B$ into $E_T$ and keep training.
>
> Secondly, if we directly train the low-rank factorization $AB$ from random initialization (as suggested by the reviewer), it will not work. This implies that the success of GrokTransfer is not due to the low-rank constraint, but rather the specific embedding being transferred. We have run the experiments for the target model with low-rank embedding and both A,B are randomly initialized, and show the results in **Figure 15(a)** (page 28) in the updated pdf. The config was selected by grid search the same as those in Figure 6 and 8. We randomly initialize A instead of transferring from the weaker model. We run the experiments for both MLP and transformers and find that the target model fails to perfectly generalize in 4000 epochs, while both the target model trained by GrokTransfer and trained from scratch can perfectly generalize in 4000 epochs.
>
> Thirdly, we wish to mention that $A, B$ are not necessarily low-rank matrices. In **Figure 15(c)**, we show that with full-rank $A, B$ (i.e., when $d_W=d_T$), GrokTransfer still archives the same acceleration effect. We used low-rank matrices in most of our experiments only because we wanted to show that even when the weak model has a low-dimensional embedding and cannot generalize perfectly, its embedding can still enable the target model to perfectly generalize with much faster speed.
>
> Our intuition is that initializing the embedding layer with an informative embedding would help build a good landscape when minimizing the loss function. **We wish to emphasize that low-rankness is not a necessary part of our story.**
>
> - ***The general motivation of the method appears to be applying dimensionality reduction to pre-process inputs that have very low SNR......***
>
> We’d like to clarify that the motivation is not to apply dimensionality reduction, since low-dimensionality is not a crucial aspect of our method. Instead, the motivation is to use the embedding learned in the weaker model to accelerate learning in the target model. As for why such embedding is helpful, we believe that it can differ between different problems.
>
> In the XOR problem, it is indeed true that the learned embedding is effectively performing a dimensionality reduction to improve the SNR. However, this mechanism doesn’t necessarily apply to other problems. For example, modular addition has noise-free data, and a learned embedding is still very useful for eliminating grokking. The general motivation of our method and a unifying view across different settings is that an informative embedding is crucial.
>
> - ***Method introduces an additional layer of complexity in the training pipeline, due to having to train an additional model which would require its own architectural and hyperparameter sweeps.***
>
> We agree that training and tuning an additional weak model makes the pipeline more complex. That said, we found that our method is quite robust to architectures and hyperparameters, and in our experiments it’s usually easy to find a weak model that works. The total computation cost is still much lower than training the target model from scratch.
>
> - ***Minor comments on notation...***
>
> Thank you for pointing out these issues. In Section 2.2, we use $d_W$ and $d_T$ to represent the embedding dimension of the Weak and Target model to distinguish the difference. We have clarified this in the updated pdf.
>
> For $y=x1x2$, we did mean to say $y=x1 * x2$, like $(p,2)$-parity task. We call it XOR distribution because opposite clusters share the same label and the projection of the data distribution looks like a XOR (see Figure 5a or Figure 11 middle for example).
>
> You are right that the notation p is overloaded. We will fix this in the next version.

---

> ### Author Response · Authors · 2024-11-23
> **Response to Questions**
>
> - ***Do the conditions in the theoretical analysis, under which grokking is mitigated for the XOR task, have any implications on what are the causes of grokking? For instance, it possibly suggests several causes of grokking, including the SNR, overparametrization (A5), initialization (A3), etc.***
>
> For the XOR task, SNR is the important cause of grokking. However, we don’t think this applies to other problems. Our work suggests a more general view that a good input embedding is an important factor that controls grokking across different problems. For the XOR problem, a good embedding turns out to be one that increases SNR.
>
> Most conditions in the theoretical analysis are the same as previous works [1,2,3] studying benign overfitting. Some are used to guarantee near-perfect generalization, e.g. SNR; some are technical conditions that come from concentration inequalities, e.g. (A5), and trajectory analysis (A3). Empirically, we find that grokking is neither caused by model structure nor hyperparameters such as learning rate, weight decay. Grokking is more related to the data structure (embedding), and the target function to be learned.
>
> - ***Are there failure cases of the method? I do not see mentions of them in the limitations.***
>
> Thank you for your question. A limitation of our method is that if a given problem lacks a significantly smaller model capable of achieving nontrivial generalization, GrokTransfer would not reduce computational costs and may provide limited utility. We will incorporate this discussion into the manuscript.
>
>
>
> [1] Benign Overfitting and Grokking in ReLU Networks for XOR Cluster Data.
>
> [2] Benign Overfitting in Linear Classifiers and Leaky ReLU Networks from KKT Conditions for Margin Maximization
>
> [3] Benign Overfitting without Linearity: Neural Network Classifiers Trained by Gradient Descent for Noisy Linear Data

---

> > ### Comment · Reviewer_s3cV · 2024-11-27
> >
> > Thank you for the detailed response to my review.
> >
> > The authors argue that $A$ and $B$ can be merged into the full matrix $E_T$ at the end of training, with minimal effects on the performance. The mentioned Figure 15(b) does not seem to mention what task this is evaluated on, but I assume it is similar from the ones in the original paper where parameterizing $E_T$ as $AB$ is already sufficient to fully learn the (toy) task. This makes the experiment rather self-fulfilling, since a fully parameterized $E_T$ matrix is not even necessary for the task in the first place. Instead, it would be more convincing if the authors were to demonstrate experiments on tasks where the greater expressiveness provided by $E_T$ is necessary for achieving good performance.
> >
> > Also, while the authors argue that "low-rankness is not a necessary part of our story" and that "the motivation is not to apply dimensionality reduction, since low-dimensionality is not a crucial aspect of our method". Figure 15(c) demonstrates the converse to me. Without the low rank conditions, the main contribution and strength of the method, which is providing large speed-ups in combined training time, disappears. My concern was not that $A$ and $B$ has to be low-rank for the model to converge, but rather $A$ and $B$ has to be low-rank for the method to be effective in terms of accelerating convergence.

---

> ### Author Response · Authors · 2024-11-29
>
> Thank you for your reply and for taking time to engage with our response. We believe there may be some misunderstandings regarding certain aspects of our method, particularly the source of the generalization speed-up and the interpretation of our results in Figure 15. We would like to clarify these points further to address your concerns.
>
> - ***The authors argue that A and B can be merged into the full matrix E_T  at the end of training, , with minimal effects on the performance.***
>
> We wish to clarify that we did not claim A,B can only be merged at the end of training. Instead, this merge operation can happen in an early training phase, e.g. at the 100th epoch—when the model has less than 10% test accuracy, as shown in Figure 15(b).
>
> - ***The mentioned Figure 15(b) does not seem to mention what task this is evaluated on***
>
> Thank you for pointing this out. Figure 15(b) is trained on the modular addition task, with the same weak and target model in Figure 6(a): a 2-layer MLP as the weak model and an 8-layer transformer (embedding dim=512) as the target model.
>
> - ***This makes the experiment rather self-fulfilling, since a fully parameterized E_T  matrix is not even necessary for the task in the first place. Instead, it would be more convincing if the authors were to demonstrate experiments on tasks where the greater expressiveness provided by E_T  is necessary for achieving good performance.***
>
> Even though from an expressive power point of view a low-rank embedding matrix suffices, that doesn’t mean it’s easy to find a solution that generalizes optimally. As we showed in Figure 15a, directly training low-rank factorization AB from random initialization only yields <50% test accuracy in the modular addition problem. On the other hand, with the standard fully parameterized embedding E_T, the model can eventually achieve 100% test accuracy, but only through grokking. This means that from the training point of view, if we only consider random initialization, the full parameterization is still necessary.
>
> By using an initialization transferred from a weaker model, our method GrokTransfer clearly improves over random initialization (with or without low-rank parameterization).
>
> - ***Without the low rank conditions, the main contribution and strength of the method, which is providing large speed-ups in combined training time, disappears.***
>
> We wish to clarify that **the embedding dimension of the weak model takes up only a small proportion of the total computation FLOPs** in the model and has little influence on the combined training time. Factors like model width, depth, architecture, and the use of CUDA have a more significant impact on training time. In GrokTransfer, the combined training time of the weak and target models is not drastically affected by the weak model's embedding dimension, as only a small portion of the target model's FLOPs comes from the embedding layer, whose FLOPs depend linearly on this dimension.
>
> To investigate the influence of the weak model's embedding dimension on training time, we have added ablation experiments by varying the embedding dimension of the weak model. The results are summarized in the table attached below, showing that the training time per epoch does not drastically change with the embedding dimension.
>
> ### **Wall-clock Time (ms) of the Backward Pass (Model Training) per Epoch (averaged on 100 epochs)**
> | **Embedding Dim (Weak model)** | **Weak Model** | **Target Model (GrokTransfer)** | **Target Model  (scratch)** |
> |---------------|---------------------|--------------------|------------------|
> | 4             | 0.746                 | 36.119             | 35.642            |
> | 8            | 0.702                 | 37.360             | 35.642           |
> | 128           |0.855                 | 36.724             | 35.642            |
> | 256           | 0.801                 | 37.033             | 35.642            |
> | 512           | 0.827                 | 37.650             | 35.642           |

---

### Official Review · Reviewer_uoKT · 2024-11-03

**Soundness:** 2
**Presentation:** 2
**Contribution:** 3
**Rating:** 8
**Confidence:** 3

**Summary:**

The paper introduces GrokTransfer, a method that expedites grokking by transferring embeddings from a weaker model. Through a simple XOR classification task, the authors offer both theoretical and empirical justification for GrokTransfer. Experiments on other algorithmic tasks show its effectiveness in embedding transfer from FNN to FNN/transformers.

**Strengths:**

1. The authors provide a sensible justification for the core idea of their method through a motivating experiment in Section 2.1.
2. They demonstrate its effectiveness both empirically and theoretically, highlighting improvements in computational costs and performance.
3. The proposed method is simple and straightforward, and if it can generalize to broader tasks and architectures, it has significant potential.

**Weaknesses:**

1. The authors tested three algorithmic tasks—modular addition, modular multiplication, and the (40,3)-parity task—for FNN->FNN transfers. However, they provided results for only one task (modular addition) in the FNN->transformers setting. Given that FNN->transformer transfers are more practical and high in potential, there seems to be no reason to exclude modular multiplication and the (40,3)-parity task. Without these experiments, I can't help but think that the generalizability of GrokTransfer is limited.

**Questions:**

1. Regarding W1, the FNN->transformer setting was tested on only one task. I would like to see GrokTransfer's performance on more algorithmic tasks in this setting, especially in comparison to GrokFast. According to Table 1, training is very fast with GrokTransfer, so it should be quick to perform experiments on additional tasks (including those not tested in the paper).

2. Grokking is known to occur beyond algorithmic data [1], which is already cited in the paper. I would like to see how GrokTransfer performs on non-algorithmic tasks, such as image classification with MNIST, as explored in [1].

3. A new paper [2] has been recently released on accelerating grokking through weight transfer based on the Kolmogorov-Arnold (KA) representation theorem. Your approach seems simpler, but the approach in [2] can even handle two non-standard arithmetic tasks—composition of operations and systems of equations. How does GrokTransfer compare to [2], in terms of theoretical basis, performance, and practicality?

[1] Ziming Liu, Eric J Michaud, and Max Tegmark. Omnigrok: Grokking beyond algorithmic data. In The Eleventh International Conference on Learning Representations, 2023.

[2] Yeachan Park, Minseok Kim, & Yeoneung Kim. Acceleration of Grokking in Learning Arithmetic Operations via Kolmogorov-Arnold Representation. arXiv preprint arXiv:2405.16658, 2024

Minor
- Typo at line 255: “need [to] learn”
- There are reference mistakes in the paper, concerning arXiv citations that should point to conference proceedings. Here is an example of wrong references:

Tanishq Kumar, Blake Bordelon, Samuel J Gershman, and Cengiz Pehlevan. Grokking as the transition from lazy to rich training dynamics. arXiv preprint arXiv:2310.06110, 2023.
(Published in ICLR 2024)

Note: the minor points did not influence my final score.

---

> ### Author Response · Authors · 2024-11-23
>
> We thank the reviewer for their review and address their comments below. We hope that the response satisfactorily addresses the reviewer’s questions and that the reviewer will consider raising their score in light of our response.
>
> - ***The authors tested three algorithmic tasks. ...... Without these experiments, I can't help but think that the generalizability of GrokTransfer is limited.***
>
> Thank you for your question. We have included the comparison for modular multiplication task in Figure 13 Left (page 27) in the updated pdf. As for the (40,3)-parity task, we followed the format in [1] and trained several transformers on the task, none of them showing delay generalization or grokking. Please see Figure 13 Right for the dynamics of a 2-layer and a 4-layer decoder-only transformer on (40,3)-parity task. We hypothesize that due to the existence of positional embedding, it is much easier for the transformer to identify the signal tokens than MLP, thus grokking is not observed.
>
> - ***Regarding W1, the FNN->transformer setting was tested on only one task.......***.
>
> Thank you for your suggestion. We have included the comparison with GrokFast on modular multiplication tasks in the updated pdf. As shown in Figure 16 (page 28), our method still outperforms GrokFast.
>
> - ***Grokking is known to occur beyond algorithmic data [1], which is already cited in the paper. I would like to see how GrokTransfer performs on non-algorithmic tasks, such as image classification with MNIST, as explored in [1].***
>
> Thank you for your suggestion. We have added results for the MNIST dataset in Figure 14 (page 27) in the updated pdf. Following the setting in [1], 200 samples are randomly selected as the training data. We train a three-layer MLP as the weak model and transfer its embedding to the target model, a four-layer MLP, and compare its dynamics with training from scratch, both with alpha=100. Figure 14 shows that GrokTransfer outperforms training from scratch and almost eliminates the generalization delay.
>
>
> - ***A new paper [2] has been recently released on accelerating grokking through weight transfer based on the Kolmogorov-Arnold (KA) representation theorem......***
>
> In section 4.3 in [2], it studies weight transfer across different arithmetic tasks. Specifically, it first trains a transformer on one task then transfers its embedding to another task, which is very similar to what [3] did.
>
> E.g. [3] tried to first train a transformer on modular addition till perfect generalization, and transfer the trained embedding to another transformer on modular multiplication. **In Figure 2 in [3], they found that it will actually prevent the new transformer from perfect generalization.**
>
> Given a task A, the goal of [2,3] is to construct informative embedding from different but similar tasks B,C to accelerate generalization, while our idea is to utilize existing training data of A to construct informative embedding. Their weight transfer across different tasks implies that the model is trained on many more data points than ours and needs to be trained multiple times, which is not directly comparable to our work.
>
> - ***There are reference mistakes in the paper, concerning arXiv citations that should point to conference proceedings.***
>
> Thank you for pointing out this issue. We have fixed it in the updated pdf.
>
> [1] Omnigrok: Grokking Beyond Algorithmic Data
>
> [2] Acceleration of Grokking in Learning Arithmetic Operations via Kolmogorov-Arnold Representation
>
> [3] Towards Empirical Interpretation of Internal Circuits and Properties in Grokked Transformers on Modular Polynomials

---

> ### Comment · Reviewer_uoKT · 2024-11-25
>
> Thank you for your responses to my concerns. While I appreciate the efforts to address my comments, I would like to provide some additional feedback and seek clarification on certain aspects:
>
> Training Data Size in Comparison to Omnigrok [1]
> - I note that my concern regarding the limited experiments with FNN-to-Transformer transitions has been partially addressed. I observed that your MNIST experimental setup uses only 200 training samples, whereas the Omnigrokk paper employed 1000 training samples (refer to page 5 in [1] and official code). Could you elaborate on the rationale for this decision?
>
> Presentation of Results
> - I believe the research community would find the FNN->Transformer experiments more significant than the FNN->FNN experiments. In this regard, I suggest reorganizing the presentation of results. Specifically, consider including the FNN->Transformer version of Figure 6 in the main text, as it represents a key aspect of your contribution. The FNN->FNN results, while informative, could be moved to the Appendix to streamline the narrative and focus on the most impactful findings.
>
> Highlighting Revised Sections
> - To further facilitate the review process, it would be very helpful if the sections of the manuscript that have been revised in response to reviewer comments are clearly highlighted in the PDF (e.g., using color or annotations). This will allow reviewers to quickly locate and evaluate the changes made based on their feedback.
>
> Thank you again for your thoughtful responses and for the hard work put into addressing the review comments. I would like to update my score after seeing additional responses and revisions from the authors.

---

> > ### Author Response · Authors · 2024-11-26
> >
> > Thank you for your constructive feedback.
> >
> > - ***MNIST experimental setup uses only 200 training samples***
> >
> > We have included a MNIST experiment with 1000 training samples in Figure 18 in the updated pdf to align with the setting in Omnigrok [1]. We can see that GrokTransfer outperforms training from scratch and almost eliminates the generalization delay.
> >
> > - ***Presentation of Results***
> >
> > Thank you for your suggestion. We have reorganized the manuscript and moved the original Figure 6 to Appendix and highlighted all revised parts with coloring red. Please see the updated pdf for details.

---

> > > ### Comment · Reviewer_uoKT · 2024-11-27
> > >
> > > I find that my concerns have been properly addressed. Based on the paper's theoretical and empirical contributions, I will update my score to 8.
> > >
> > > While I find the current experiments well-executed, I believe the impact of the paper could be further enhanced by extending the experiments to more challenging image classification datasets (with more data samples greater than 1000), such as CIFAR-100, STL-10, or even ImageNet. If the paper is accepted, I encourage the authors to consider conducting such experiments for the camera-ready version. These additional experiments would add significant value to the research, further solidifying its contribution to the field.

---

### Official Review · Reviewer_opL6 · 2024-11-06

**Soundness:** 3
**Presentation:** 4
**Contribution:** 2
**Rating:** 6
**Confidence:** 3

**Summary:**

This paper investigates "grokking," a phenomenon where neural networks initially struggle to generalize, only to suddenly achieve near-perfect generalization after extended training. To accelerate generalization, the authors propose GrokTransfer. GrokTransfer leverages the embedding from a smaller, preliminary model that achieves moderate test performance, transferring it to a stronger target model to speed up generalization. This approach successfully eliminates delayed generalization on a synthetic XOR task and performs well across various algorithmic tasks on both fully connected networks and Transformers. The results suggest that GrokTransfer could reshape training dynamics, enabling models to generalize without delay.

**Strengths:**

- The proposed method is interesting and novel to my knowledge. It first trains an under-parameterized model that is incapable of perfectly interpolating the data, and then uses the learned data embedding to facilitate the grokking of an over-parameterized model. It’s like approaching a problem by first crafting a simple but general solution that works well for most cases and then refining the solution to cover all the rest of the cases. Intuitively, this eliminates a lot of unnecessary competition among various not-so-simple solutions in an over-parameterized model, hence accelerating the learning process.
- The paper is very well written. It includes an extensive discussion of the related work, a clear presentation of the motivation behind the proposed method, and a detailed case study that explains how and why the method works.

**Weaknesses:**

- It is unclear how well the method would generalize and scale to more complicated problems where such acceleration can make a real impact. The algorithmic problems are useful for analysis but are too simple in comparison with real-world problems which often involve high-dimensional inputs. For high-dimensional inputs with many redundant features, it is likely for the weaker model to lock onto degenerate solutions that would hinder the stronger model from grokking. It would be interesting to see if this is really the case or not.
- The paper does not provide much insight on how to design or choose the weaker model. Clearly, the weaker model can neither be too weak nor too strong, i.e. there is a trade-off. If it is too weak, the solution may degenerate and thus make it harder for the stronger model to grok. If the weaker model is too strong, then it may take too much time to train the weaker model. Therefore, for this method to be truly useful, there should be some general rule of thumb for choosing the weaker model, otherwise, much time could be wasted on trial and error.
- As the authors have mentioned in the paper, the theoretical result only considers a relatively simple XOR task. There does not seem to be any clear indication if the analysis could potentially be applied to more general problems. Therefore, the significance of this result is in doubt.

**Questions:**

- Would a “smoother” version of the proposed method work even better? For example, one could start from a single embedding layer with a small number of neurons and gradually deepen/widen it, i.e., adding more layers and adding more neurons to each layer.

---

> ### Author Response · Authors · 2024-11-23
>
> We thank the reviewer for the detailed review and for appreciating the contributions of our paper. It is encouraging to see that the reviewer thinks our method is “interesting” and “novel” and has a detailed case study.
>
> - ***It is unclear how well the method would generalize and scale to more complicated problems where such acceleration can make a real impact. ......  It would be interesting to see if this is really the case or not.***
>
> Thank you for your suggestion. We have added results for the MNIST dataset in Figure 14 (page 27) in the updated pdf, where grokking was observed with large initialization in [1]. Following the setting in [1], 200 samples are randomly selected as the training data. We train a three-layer MLP as the weak model and transfer its embedding to the target model, a four-layer MLP, and compare its dynamics with training from scratch, both with initialization scale $\alpha$=100. Figure 14 shows that GrokTransfer outperforms training from scratch and almost eliminates the generalization delay.
>
> - ***The paper does not provide much insight on how to design or choose the weaker model. ......***
>
> Thanks for the comment. The general rule of thumb we found is to start from a computationally light model (e.g. a small MLP) as the weak model and see if it can show some degree of generalization. If not, we can gradually grow the model to a larger size until it shows nontrivial generalization.
>
> Take modular arithmetic tasks as an example. Empirically it’s sufficient for the target model to perfectly generalize if the weak model can achieve around 40% validation accuracy on modular arithmetic tasks, and a two-layer MLP with 4-dim embedding can already learn a good embedding with periodic structure. See Figure 12 (page 27) in the updated pdf. The t-SNE embedding shows the embedding has structure similar to Fourier embedding with frequency 1/66.
>
> - ***As the authors have mentioned in the paper, the theoretical result only considers a relatively simple XOR task. There does not seem to be any clear indication if the analysis could potentially be applied to more general problems. Therefore, the significance of this result is in doubt.***
>
> We agree that our theoretical result is in a simple setting and we do not think that the same analysis applies to more general problems. Note that rigorously analyzing the training dynamics in neural networks is a big theoretical challenge and even simple settings like ours are already difficult to analyze. Our theoretical result is meant to provide some initial justification and insights about how our algorithm can possibly work, especially when the weaker model doesn’t generalize optimally.
>
> - ***Would a “smoother” version of the proposed method work even better? For example, one could start from a single embedding layer with a small number of neurons and gradually deepen/widen it, i.e., adding more layers and adding more neurons to each layer.***
>
> We appreciate the reviewer’s question and suggestion. We think this is an interesting idea. For the purpose of accelerating grokking, we found that transferring from a single weaker model to a target model is already sufficient. We agree that the proposed “smoother” version is potentially useful and is worth studying in future work.
>
> [1] Omnigrok: Grokking Beyond Algorithmic Data

---

> ### Comment · Reviewer_opL6 · 2024-11-26
>
> I thank the authors for carefully addressing my concerns. The MNIST experiment shows some promising signs of the scalability of the method, which is good, but I was thinking about something more realistic and challenging (e.g., ImageNet; natural language modeling). I understand that such experiments may be too much to ask during rebuttal, but still, it would be a big plus if the authors could show that their method works at a scale and complexity closer to real-world applications. Maybe this can be done in the future. For a preliminary study, the current result seems to suffice.
>
> For the second point, it makes perfect sense to start from a computationally light model and gradually grow the model to a larger size until it shows nontrivial generalization. However, a key issue lies in the notion of "nontrivial generalization". For modular arithmetic tasks, 40% validation accuracy is sufficient, but this is only known after training the target model. For other tasks, does 40% accuracy also suffice? Could some require more, let's say, 80% accuracy? I think the paper would benefit from more empirical analysis of the relation between the accuracy of the weak model and the time required to train the target model.

---

### Official Review · Reviewer_g3Q3 · 2024-11-08

**Soundness:** 3
**Presentation:** 3
**Contribution:** 2
**Rating:** 3
**Confidence:** 3

**Summary:**

This paper proposes a method to accelerate grokking via embedding transfer from a week model. The authors first observe that data embedding plays a crucial role in determining whether generalization is delayed. Then they initialize the embedding weights extracted from  training a smaller and weaker model to the target, strong model. Finally, the authors give rigorous proof on a synthetic XOR task with simple network settings, and provide empirical studies showing the effectiveness of the proposed GrokTransfer method on fully-connected network and transformers.

**Strengths:**

1) This paper is clearly written and the idea is easy to follow.
2) The choice of the task (XOR) and the setting of the model (very simple network) is easy to interpret for the purpose of the finding of this paper.

**Weaknesses:**

1) the key observation which is that the data embedding plays a crucial role in the generalization delay is not well studied. As the proposed accelerating method is based on this observation, it would be better to provide more evidence to say this observation is indeed the correct one. In addition to the modular addition task, providing more results on other tasks would make this observation more convincing. Also, instead of data embedding layer only, is the initialization of other layers a possible reason to cause the delay? An ablation study on other possible reasons to cause the delay would be better.

2) In addition to simple settings, verification on more complicated tasks would make the finding more general and solid.

3) In the theoretical analysis, the proof seems a little bit limited because the choice of very simple setting.

4) It would be better if the authors make a clearer study on how to choose the "weaker and smaller" model to get the acceleration with good test performance.

**Questions:**

1) In section 3.2, could you add more details on getting the SNR with different embeddings?

2) The intuition on why the data embedding  make a difference on the generalization speed is still not very clear.

If we take all layers beyond the last linear layer (which includes data embedding layer and some hidden layer) as doing feature engineering, as mentioned in [1], will this claim (in section 3.2, ``with this new embedding, data points are well separated in a three-dimensional space with a relatively high signal-to-noise ratio (SNR) compared to the original embedding'') still hold if we pass this data embedding into above hidden layers? Does the separation among the feature vectors got from the last hidden layer make more sense than the one among data embeddings?



[1] Papyan, V., Han, X. Y., & Donoho, D. L. (2020). Prevalence of neural collapse during the terminal phase of deep learning training. Proceedings of the National Academy of Sciences, 117(40), 24652-24663.

---

> ### Author Response · Authors · 2024-11-23
> **Response to Weaknesses**
>
> We thank the reviewer for their review and address their comments below. We hope that the response satisfactorily addresses the reviewer’s questions and that the reviewer will consider raising their score in light of our response.
>
> - ***the key observation which is that the data embedding plays a crucial role ...***
>
> Thank you for the comment. In section 2, we show that good data embedding can effectively change the training dynamics and cause the grokking phenomenon to disappear. This serves as a motivation to propose our GrokTransfer method. The observation that data embedding plays an important role also appeared in [2], where the authors found that generalization coincides with the emergence of periodic structure in the embedding (Figure 1 in [2]).
>
> On the other hand, we do not claim that data embedding is the only reason behind grokking, or that changing the data embedding is the only way to accelerate grokking. We think it is possible that other factors, such as initialization of other layers, also play a role here. But we focus on the embedding layer because it is easy to transfer the embedding across different model sizes and different model architectures, making it more broadly applicable. We think the fact that GrokTransfer works so well across different tasks and different architectures is convincing evidence that changing the data embedding alone is sufficient for eliminating grokking.
>
> - ***In addition to simple settings, verification on more complicated tasks would make the finding more general and solid.***
>
> Thank you for your suggestion. We have added results for the MNIST dataset in Figure 14 (page 27) in the updated pdf, where grokking was observed with large initialization scale in [1]. Following the setting in [1], 200 samples are randomly selected as the training data. We train a three-layer MLP as the weak model and transfer its embedding to the target model, a four-layer MLP, and compare its dynamics with training from scratch, both with alpha=100. Figure 14 shows that GrokTransfer outperforms training from scratch and almost eliminates the generalization delay.
>
> - ***In the theoretical analysis, the proof seems a little bit limited because of the choice of very simple setting.***
>
> We agree that our theoretical result is in a simple setting. Note that rigorously analyzing the training dynamics in neural networks is a notoriously difficult theoretical challenge and even simple settings like ours are already difficult to analyze.
>
> Our theoretical result is meant to provide some justification and insights about why our algorithm works. A more comprehensive theoretical justification is beyond the scope of this paper.
>
>
> - ***It would be better if the authors make a clearer study on how to choose the "weaker and smaller" model to get the acceleration with good test performance.***
>
> Thank you for the suggestion. Empirically, we can start from a computationally light model as the weak model and see if it can show some degree of generalization. If not, we can gradually grow the model size until the model shows nontrivial generalization. We will add this discussion to the manuscript.

---

> ### Author Response · Authors · 2024-11-23
> **Response to Questions**
>
> - ***In section 3.2, could you add more details on getting the SNR with different embeddings?***
>
> In the original embedding, each sample x is a concatenation of signal and noise vectors. The SNR is the ratio of the norm of the signal vector and the norm of noise vector, i.e. \sqrt(2/(\epsilon (p-2))). After training the weak model, we extract its first linear layer W, and apply a linear transformation to the data distribution with W, i.e. x -> W^T x. Wx can be interpreted as the new embedding for x. Since the signals are the first two elements in x, and W has learned this structure, the SNR of W^T x will be larger.
>
> - ***The intuition on why the data embedding make a difference on the generalization speed is still not very clear.***
>
> In the XOR task, the learned data embedding projects the original representation to a 3D space with higher SNR, enabling the model to generalize without delay. In the modular addition task, the learned data embedding has a structure similar to a Fourier embedding, which effectively changes the loss landscape.
>
>
> - ***If we take all layers beyond the last linear layer as doing feature engineering, as mentioned in [1], will this claim...***
>
> It’s a good question. It’s widely believed that the low-level layers learn fundamental features while the deeper layers learn more complex and task-specific features [3]. Since the weak model may not generalize optimally, we tend to believe it does not learn all the necessary features or may have learned some incorrect features. To minimize the misinformation to be transferred into the target model, we think only transferring the data embedding is a good idea.
>
> As for the reviewer’s question about whether the claim in Section 3.2 holds for later layers, we note that this claim only applies to the one-hidden-layer setting studied in Section 3, where the hidden layer we look at is already the penultimate layer. We agree that further investigating this for deeper networks is an interesting direction, but it’s beyond the scope of this work.
>
> [1] Omnigrok: Grokking Beyond Algorithmic Data
>
> [2] Towards Understanding Grokking: An Effective Theory of Representation Learning
>
> [3] Visualizing and Understanding Convolutional Networks

---

> ### Author Response · Authors · 2024-12-03
>
> We thank the reviewer for their review. We would greatly value the opportunity to discuss and remain available to address any additional questions or concerns during the discussion period. In light of the clarifications, revisions, and additional evaluations we have provided, we kindly encourage the reviewer to revisit their assessment and consider adjusting their final rating before the discussion period concludes.

---

### Official Review · Reviewer_GjdD · 2024-11-09

**Soundness:** 3
**Presentation:** 2
**Contribution:** 3
**Rating:** 5
**Confidence:** 4

**Summary:**

The paper studies ways to reduce the "delayed generalization" in grokking, by first learning embeddings in a smaller, faster model where generalization isn't delayed, and then transferring the embeddings to a larger model. This is shown empirically in various examples including modular addition, multiplication and parity with fully-connected nets or transformers. The authors also show theoretically that fast generalization occurs in XOR when applying the GrokTransfer technique on top of a weak solution that was found with only 3 neurons (this solution is shown to be found empirically).

**Strengths:**

Grokking is a surprising phenomenon that leads to seemingly unpredictable outcomes. The proposed GrokTransfer method seems like an empirically effective to mitigate this. The theoretical claims also justify why good embeddings obtained from a small model can help accelerate generalization in a larger model. Overall, this makes the work interesting and significant.

**Weaknesses:**

* The theory on XOR would benefit from a better understanding of the "weaker model", which is currently mostly empirical. It would great to have even partial results for this part, even in a simpler setting (e.g. training a single neuron on this data might be tractable and similar to previous work?)

* The presentation could be improved in various ways, in particular the notion of "small model" considered isn't very precisely defined and would benefit from clarification. Other aspects that could be discussed further include trade-offs between width and iterations needed to generalize: it seems that smaller models reduced the "delayed generalization" but also slow down convergence in terms of epochs. Perhaps this changes when looking at "compute/flops", and it would be good to include such plots.

See also the questions below.

**Questions:**

* Could you comment on why you chose a weak model with 3 neurons in Section 3.2? Would the story change if you had 4 of them and achieve perfect accuracy? My impression is that the delayed generalization mostly happens when you are heavily over-parameterized and can very quickly memorize all the training data in early epochs, which still wouldn't happen with 4 neurons. Also, would the transfer still work if the weak model had fewer than 3 neurons?

* For the modular addition/multiplication problems, what would be the smallest weak model that can lead to good transferrable embeddings (in practice, but also in theory with an appropriate construction?)

* For Theorem 3.2, is this in a kernel regime? Would training only the second layer instead give similar guarantees? Regardless, a brief discussion after the statement would be a good addition.

Other, minor:
* Figure 4: Should I infer from the caption that the test accuracy becomes high around the same point (near 100 epochs), while the larger model also displays the property that training accuracy is high very quickly? Also, is it reasonable to say that the small model does not exhibit grokking here? Please clarify these points since they seem important for the story.
* L 298: Do you mean that whenever the test accuracy is around 75% or above, you found empirically that the neurons consist of three such features? Please rephrase as this is not very clear.

---

> ### Author Response · Authors · 2024-11-23
> **Response to Weaknesses**
>
> We thank the reviewer for the detailed review. It is encouraging that the reviewer thinks our work is interesting and significant. We hope that our response satisfactorily addresses the reviewer’s questions/concerns.
>
> - ***The theory on XOR would benefit from a better understanding of the "weaker model", ...... e.g. training a single neuron on this data might be tractable and similar to previous work?)***
>
> We appreciate the question regarding the theoretical analysis of the weak model. In the single neuron case, the trajectory analysis is similar to that in [1]. Specifically, the neuron will gradually align with one of the four feature directions (the direction with the most samples in the training data) and will only be able to learn that feature due to model expressivity. Consequently, the test accuracy will always be around 25% and the training accuracy never exceeds 50%, which is verified empirically in Figure 10 Left and Middle (page 26) in the updated manuscript. The two or three-neuron cases are significantly more challenging to analyze and are beyond the current toolkit in deep learning theory, mainly due to the complex interactions between different neurons. Another type of setting we know how to analyze is when the number of neurons is sufficiently large (e.g., at least logarithmic on the sample size); we did not focus on such settings because we wanted to highlight that the weaker model doesn’t have to generalize perfectly.
>
>
> - ***The presentation could be improved in various ways, in particular the notion of "small model" considered isn't very precisely ...... Perhaps this changes when looking at "compute/flops", and it would be good to include such plots.***
>
> Thank you for your suggestion. Here “smaller model” generally refers to a model with relatively smaller expressivity than the target model. This is similar to what has been studied in the contexts of distillation and weak-to-strong generalization. We have updated the pdf to include this clarification.
>
> The number of epochs needed by the weak model to generalize does increase compared to that of the target model. However, the wall-clock time has a significant decrease compared to training the target model from scratch. In Table 1 (page 10), we see that the wall-clock time used to train the weak model plus that to train the target model with GrokTransfer is much smaller than that of training the target model from scratch, thus boosting the generalization speed in terms of time. We didn’t include compute/flops because it is very hard to measure the flops for transformers in the backward pass when using CUDA. For the forward pass, we follow the formula in Table 1 in [4] and calculate the flops for Figure 8 Left. The flops of GrokTransfer is around 1e10, and the flops of training from scratch is around 1e11. The flops of training the weak model is around 1e7, which is negligible compared to the flops of training the target model. Please see more details in Section A.4 in the updated pdf.

---

> ### Author Response · Authors · 2024-11-23
> **Response to Questions**
>
> - ***Could you comment on why you chose a weak model with 3 neurons......***
>
> We use 3 neurons because we want to show that even when the weaker model doesn’t generalize perfectly, its embedding can still help the target model to achieve optimal generalization with much faster speed, which matches our empirical observation. If the weaker model has 4 (or more) neurons and achieves perfect generalization itself, the story won’t change and the target model can still generalize after one step with GrokTransfer.
>
> If the weaker model has two neurons, GrokTransfer still works and our theory still applies. The weak model will learn two out of four features and its embedding will project the data onto a 2-D space, as shown in Figure 11 Middle (page 26) in the updated version.
>
> If the weak model only has one neuron, GrokTransfer does not work. As shown in Figure 11 Left (page 26), the weak model’s embedding projects the data onto a 1-D space, where both the +1 and -1 classes are located on both sides of the original point 0, making it impossible for the target model to classify them. Please see more details on page 25.
>
>  - ***For the modular addition/multiplication problems, what would be the smallest weak model...***
>
> We appreciate the reviewer’s question. Empirically we find that it’s sufficient for the target model to perfectly generalize as long as the weak model can achieve a nontrivial test accuracy (e.g. 40%) on modular arithmetic tasks. For example, a two-layer MLP with 4-dim embedding is sufficient. We also found that such a small model can already learn a good embedding with a periodic structure. See Figure 12 (page 27) in the updated pdf. The t-SNE embedding shows the embedding has a structure similar to Fourier embedding with a frequency 1/66.
>
> Theoretically, we believe that as long as the model has enough expressivity to extract useful features, it can be used as a weak model. [5,6] both give constructions for 2-layer MLPs on the modular addition task based on the Fourier embedding.
>
>
> - ***For Theorem 3.2, is this in a kernel regime?...***
>
> Thank you for your question. It’s not in the kernel regime but instead a feature learning regime, where the learning rate is much larger than the initialization scale. In this task, only training the second layer cannot achieve one-step generalization. We have updated the manuscript to clarify this point.
>
> - ***Figure 4: Should I infer from the caption that the test accuracy becomes high...***
>
> In Figure 4, it’s correct that the test accuracy becomes high around the same point for both smaller and larger models, while the larger model’s training accuracy becomes perfect more quickly. However, note that the smaller model is much more efficient to train than the larger model due to their size difference. Our main point is that this smaller model’s embedding can be used to significantly accelerate larger model’s generalization (i.e., generalizes perfectly in 1 step), and therefore using GrokTransfer can achieve an overall speedup compared with training the larger model from scratch.
>
> As for whether or not the smaller model exhibits grokking, we agree that it isn’t as clear as the larger model. The delayed generalization phenomenon is still quite clear here, but the gap between training and test curves isn’t as pronounced. That said, this has no effect on our story, since we only care about using the smaller model to help accelerate grokking in the larger model, and so the training dynamics of the weak model doesn’t play an important role in the story.
>
> - ***L 298: Do you mean that whenever the test accuracy is around 75% or above, ...***
>
> Yes. Empirically we found that the 3-neuron NN can always achieve ~75% test accuracy as long as it is trained for sufficiently many steps and each time it learns three out of four features [\pm 1, \pm 1]. The specific features they learned depend on its weight initialization and the sampling of training data. We have clarified it in the updated pdf.
>
>
>
> [1] Benign Overfitting and Grokking in ReLU Networks for XOR Cluster Data.
>
> [2] SGD Finds then Tunes Features in Two-Layer Neural Networks with Near-Optimal Sample Complexity: A Case Study in the XOR problem
>
> [3] Benign Overfitting in Two-Layer ReLU Convolutional Neural Networks for XOR Data
>
> [4] Scaling Laws for Neural Language Models
>
> [5] Feature emergence via margin maximization: case studies in algebraic tasks
>
> [6] Grokking modular arithmetic

---

> ### Author Response · Authors · 2024-12-03
>
> We thank the reviewer for their review. We would greatly value the opportunity to discuss and remain available to address any additional questions or concerns during the discussion period. In light of the clarifications, revisions, and additional evaluations we have provided, we kindly encourage the reviewer to revisit their assessment and consider adjusting their final rating before the discussion period concludes.

---

### Author Response · Authors · 2024-11-23
**Global Response**

We appreciate the reviewers' feedback! In this response, we summarize our additional experiments, which have been incorporated into the updated PDF.

- **Section A.4.1**:
  We include an estimation of the FLOPs required for the forward pass in the weak and target models within the FNN-to-Transformer scenario (Figure 8, Left). This demonstrates that the FLOPs of the weak model can be significantly smaller than those of the target model.

- **Section A.4.2**:
  We present additional experiments on XOR cluster data, the modular multiplication task, and image classification on MNIST, where grokking was observed with large initialization scales [1].

  - For the **XOR cluster data** case study, we analyzed the minimum number of neurons required in the weak model to ensure near-perfect generalization for the target model (Figures 10, 11).

  - For the **modular addition task**, we visualized the weak model’s embedding and observed a periodic structure resembling Fourier embeddings (Figure 12).

  - For the **FNN-to-Transformer scenario**, we added experiments on modular multiplication and compared the results with GrokFast (Figures 13 Left, 16).

- **Figure 14**: We applied GrokTransfer to the MNIST dataset and observed an improvement in generalization speed.

- **Figure 15**:  We compared the dynamics of three configurations: randomly initializing both models $A$ and $B$, merging $A$ and $B$ into $A\times B$, and setting the embedding dimension of the weak model equal to that of the target model. These comparisons highlight that the success of our method is not from the low-rankness of the embedding layer but from the information encoded in the weak model's embedding.

[1] Omnigrok: Grokking Beyond Algorithmic Data

---

### Meta-Review · Area_Chair_wTJP · 2024-12-22

**Metareview:**

This paper proposes Groktransfer, a new method to accelerate groking based on transferring embeddings from a weak model. The authors provide theoretical analysis on a simple XOR classification task and numerous empirical experiments on modular addition, multiplication, and additional results on mnist in the rebuttal with different architectures of weak models and target models. There were some concerns regarding the implementation to transfer embedding matrix (low rank, full rank of A, B) from weak model to target model, and it seems addressed by authors' additional results (Fig 15). The authors should make this part clear in the main text (that not requiring the low rank of A, B) and move Fig 15 to main draft. Given the theoretical and empirical contributions of this work (novelty and superior performance), I recommend to accept this paper. The authors are urged to release the code for reproducibility and further improve the paper by including discussions/clarification paragraphs and in-depth comparison with baselines on more datasets based on reviewers questions and comments in the rebuttal period.

**Additional Comments On Reviewer Discussion:**

There are some shared concerns from the reviewers regarding the details and applicability of the proposed methods (to more tasks) and the authors addressed the concerns by providing additional experiments in the rebuttal, including more in-depth study on XOR cluster data, modular additional task, and FNN-t-Transformer scenario, and mnist experiments. The authors also strength the paper by comparing with a recent method Grokfast to show it has faster speed for groking.

There are also some questions regarding the construction of embedding matrix in Groktransfer and the authors provided additional results showing that the rank of embedding matrix in the weak model do not have specific requirement, and the crucial point is to initialize the target model's embedding matrix from the weak models'.

---

### Decision · Program_Chairs · 2025-01-22

Accept (Poster)